# A Deep Learning-Based Innovative Technique for Phishing Detection in Modern Security with Uniform Resource Locators

**DOI:** 10.3390/s23094403

**Published:** 2023-04-30

**Authors:** Eman Abdullah Aldakheel, Mohammed Zakariah, Ghada Abdalaziz Gashgari, Fahdah A. Almarshad, Abdullah I. A. Alzahrani

**Affiliations:** 1Department of Computer Sciences, College of Computer and Information Sciences, Princess Nourah bint Abdulrahman University, Riyadh 11671, Saudi Arabia; 2Department of Computer Science, College of Computer and Information Science, King Saud University, Riyadh 12372, Saudi Arabia; 3Department of Cybersecurity, College of Computer Science and Engineering, University of Jeddah, Ar Rabwah Jeddah 23449, Saudi Arabia; 4Department of Information Systems, College of Computer Engineering and Sciences, Prince Sattam Bin Abdul-Aziz University, Al Kharj 11942, Saudi Arabia; 5Department of Computer Science, College of Science and Humanities in Al Quwaiiyah, Shaqra University, Shaqra 11961, Saudi Arabia

**Keywords:** phishing detection system, deep learning, convolutional neural network, PhishTank data set, URL analysis, machine-learning

## Abstract

Organizations and individuals worldwide are becoming increasingly vulnerable to cyberattacks as phishing continues to grow and the number of phishing websites grows. As a result, improved cyber defense necessitates more effective phishing detection (PD). In this paper, we introduce a novel method for detecting phishing sites with high accuracy. Our approach utilizes a Convolution Neural Network (CNN)-based model for precise classification that effectively distinguishes legitimate websites from phishing websites. We evaluate the performance of our model on the PhishTank dataset, which is a widely used dataset for detecting phishing websites based solely on Uniform Resource Locators (URL) features. Our approach presents a unique contribution to the field of phishing detection by achieving high accuracy rates and outperforming previous state-of-the-art models. Experiment results revealed that our proposed method performs well in terms of accuracy and its false-positive rate. We created a real data set by crawling 10,000 phishing URLs from PhishTank and 10,000 legitimate websites and then ran experiments using standard evaluation metrics on the data sets. This approach is founded on integrated and deep learning (DL). The CNN-based model can distinguish phishing websites from legitimate websites with a high degree of accuracy. When binary-categorical loss and the Adam optimizer are used, the accuracy of the k-nearest neighbors (KNN), Natural Language Processing (NLP), Recurrent Neural Network (RNN), and Random Forest (RF) models is 87%, 97.98%, 97.4% and 94.26%, respectively, in contrast to previous publications. Our model outperformed previous works due to several factors, including the use of more layers and larger training sizes, and the extraction of additional features from the PhishTank dataset. Specifically, our proposed model comprises seven layers, starting with the input layer and progressing to the seventh, which incorporates a layer with pooling, convolutional, linear 1 and 2, and linear six layers as the output layers. These design choices contribute to the high accuracy of our model, which achieved a 98.77% accuracy rate.

## 1. Introduction

Phishing is a type of fraud involving technological and social approaches to collect financial and personal data from clients [1]. One of the most known techniques is the act of creating emails with a fake originator address which is known as email spoofing. Social media platforms deploy spoof emails from legitimate organizations and agencies to entice customers to browse fraudulent websites and divulge security information such as usernames and passwords. Hackers frequently use systems to intercept consumer internet account usernames and passwords, installing malicious software on machines to obtain credentials [2]. Phishers obtain user information through various means, including email, forum postings, URLs, instant chats, text messages, and phone calls [3]. Phishing content resembles legitimate content in structure, enticing consumers to browse it to obtain sensitive information [4]. Phishing’s primary goal is to gather confidential data to obtain money or commit identity theft. Phishing assaults are causing havoc for organizations all around the world [5]. The Anti-Phishing Working Group (APWG) is a nonprofit that accumulates, analyzes and distributes a list of authenticated URLs [6]. It offers progress reports on global phishing activities. From 2021 to 2022 the number of phishing attacks nearly doubled. Over 280,000 phishing attempts were reported in July 2022 [7,8,9,10,11,12]. Webmail is still a popular phishing target. The monthly number of phishing attacks on well-known brands has increased from 500 in July to 900 in November 2022. State laws exist to punish criminals who conduct phishing attacks. Under the Anti-phishing Act of 2005 [9,13] anyone involved in phishing attacks faces imprisonment for a maximum of five years and fines or both. California has the most stringent anti-phishing legislation.

To obtain confidential information, criminals create illegal reproductions of legitimate websites and emails, typically from financial institutions or other organizations that deal with financial data. This email contains logos and slogans from a reputable company. To catch users off guard, phishers send “spooled” emails to many people. Users who open these emails are usually taken to a bogus website rather than the actual company [11]. As a result, there is a significant risk of user data being misused. For these reasons, phishing is highly urgent, complex, and critical in today’s society. However, most organizations lack appropriate anti-phishing technologies to detect fraudulent URLs and protect users. If malicious code is installed on the website, installed malware and hackers may steal user information posing a significant threat to cybersecurity and user privacy. DL techniques can quickly identify malicious URLs on the internet [14,15].

Conventional URL detection relies on a blacklist established from user feedback or expert judgment [16]. The blacklist is used to validate a URL, and the URL in the blacklist is sometimes revised. In contrast, the number of harmful URLs not on the blacklist continuously grows. For example, fraudsters can use a Domain Generation Algorithm (DGA) to generate new malicious URLs to circumvent the blacklist [17]. As a result, using a comprehensive blacklist to identify malicious URLs is nearly impossible, and existing methods are incapable of detecting new dangerous URLs.

Recent PD research has concentrated on DL techniques such as Bayesian Additive Regression Trees (BART) and Graph Convolutional Network (GCN) for attribute detection in observed data sets [18]. These studies mainly focused on URL identifiers, with a few exceptions in searching for email text [19]. Numerous previous studies have concentrated on email text or URL data [20,21,22,23]. It opens the door to investigating and creating a comprehensive model that combines numerous methods on different parts of phishing scams, such as attachments, URLs, senders, images, and body text to detect phishing attempts effectively. This learning incorporates several modeling approaches to detect sophisticated phishing attacks, including boosting and bagging methods. Preprocessing text in URLs and email bodies is one of the most challenging aspects of PD. For phishing classification, XGBoost [24] controls massive databases for the text preprocessing stage, derivates essential features, and correctly handles noise. To classify malicious emails, XGBoost analyzes the embedded email body, URLs, email attachments, sender information, and other email metadata.

Modern processes require more conventional processes and human involvement, rendering them ineffective for quickly identifying malicious activities. Furthermore, according to Reddy et al. [25], these methodologies enable an attacker to prevent restrained rules and filters. Such problems can be solved by using robust chronological data sets to create a model, which lessens the need for manual PD inputs. There are several types of modeling techniques [26,27,28,29]. For example, RNN [30], Feed Forward Neural Network (FFNN) [31], Ensemble Neural Network (ENN) [32], and Artificial Neural Network (ANN) [33] are critical NN models for email detection and phishing websites [34].

Figure 1 depicts the various DL and Machine Learning (ML) approaches for identifying phishing. There are four types of detection methods: list-based, heuristic-based, ML-based, and DL-based detection.

CNN modeling approaches can manage some of the more challenging problems of new and advanced PD. It is a densely integrated ANN that can also identify images by scanning images and writing data. It comprises multiple layers that are either max-pooling, coevolutionary, or fully connected [35]. Images with coevolutionary layers can detect chrematistic characteristics [35]. By analyzing URLs, these layers can aid in the detection of phishing attacks. By including more embedded layers, the model improves detection.

The suggested framework takes URL fragments as input and uses a prescribed feature vocabulary to recognize each character in the training set. This matrix is fed into the CNN model. The enhanced CNN model is then used here to extract representative multilevel classification models. The following are a few of the key contributors to the journal.

This study proposes a method for quickly and efficiently detecting phishing websites using URL features. This technique is based on integration and DL.We created an actual data set by indexing 10,000 valid websites and 10,000 phishing URLs using PhishTank, and we evaluated them using SD metrics.The procedure of identifying phishing websites with the ensemble approach and ML is described, and the structured data set is thoroughly evaluated. Finally, the results of our testing reveal how our presented system worked adequately concerning both FPR and accuracy.

The remainder of the paper is structured as follows: Section 2 goes over previous research and focuses on data set collection, Section 3 discusses methodology, Section 4 discusses results and analysis, Section 5 confers discussion and Section 6 discusses conclusion.

## 2. Literature Review

Many academics have researched the results of phishing websites. Our approach makes use of critical concepts from prior findings. We examined previous initiatives that used URL attributes to detect phishing, which impacted our present method.

Aljofey et al. [13] described a strategy for identifying phishing relying on the URL of this publication. The researchers used multiple algorithms to evaluate the URLs of a couple of distinct data points using different types of DL approaches and hierarchical structures to compare the findings. The first method checks several URL attributes; the second investigates the website’s legitimacy by reviewing where it is featured and who administers it and the next investigates the website’s illustration appearance. DL algorithms and techniques are applied to examine the many aspects of web pages and URLs. Yao et al. [36] created a unique method for identifying phishing websites that concentrate on analyzing a URL, which was described as a precise and effective detection technique. We have divided our new NN structure into multiple concurrent parts to give you a better idea. One approach is to start by removing shallow URL attributes.

However, in [10,36], the researchers generate accurate and trustworthy deep features of URLs while using simplistic features to evaluate URL legitimacy. The maximum throughput for this technology is determined by combining the results of all segments. Thorough research on an internet data set demonstrates that our system can keep up with other detection algorithms while spending an appropriate amount of time detecting phishing websites.

Korkmaz et al. [11] proposed a method for identifying phishing webpages based on Hypertext Markup Language (HTML) Tag Distribution Language (HTDL) and URL attributes. They also constructed concise HTDL and URL properties that allowed them to create HTDL string-embedding functions without relying on third-party infrastructure, allowing one approach to work in a legitimate recognition app. Researchers tested their technique on a legitimate database of over 30,000 HTDL and URL characteristics. According to the authors, their arrangement obtained 98.24% accuracy, a 3.99% True Positive (TP) rate, and a 1.74% False Negative (FN) rate [11]. To detect zero-day phishing schemes, Stokes et al. [14] suggested a unique approach for intelligent PD based on website text properties. The researchers used the principles of consistent resource identification and sequence strategies that usually match their framework. According to the researchers, the suggested technique successfully detects phishing and zero-day attacks, with a TP rate of 95.38%. Previous studies used website text structures to create PD frameworks. Meanwhile, phishers avoided detection by using content from other websites.

In their research on a method for spoofing site forecast that uses hyperlinks as a data feed for DL models, Yerima et al. [15] looked into the efficacy of the long short-term memory (LSTM) classifier. The researcher compares an RF classifier-based method against a new RNN-based method. They performed their scientific study of web addresses using fourteen criteria. First, researchers built a model using LSTM that accepts a hyperlink as a sequence of text as input and identifies whether the URL is legitimate or fraudulent. Despite the lack of specialist expertise required to create the features, users discovered that the LSTM algorithm outperforms the RF classifier in terms of average accuracy rate. Though without preprocessing and feature formation, their approach achieves 97.4% accuracy [15]. Their research, moreover, was restricted to the text-feature perspective of webpages. Integrating other elements such as frame features and website images might enhance the efficiency of their model.

Selamat et al. [12] evaluated two URL data sets for phishing detection using logistic regression with CNN and CNN-LSTM. They gathered a data set from different sources: malware domains, malware domain lists, and phishing domains from OpenPhish and PhishTank. The database comprises over seventy thousand URLs for training and over 60,000 URLs for testing. They used the data set to train the CNN-LSTM and CNN phishing URL detection models. The LSTM approach has been selected because it acknowledges the actual data web address as input. In their trial, the CNN-LSTM architecture outperformed the other framework, obtaining an accuracy rate of about 97% for URL categorization [19]. In contrast, the suggested approach merely uses text-based characteristics and may be improved by adding additional attributes and optimizing the variables for higher accuracy. Consequently, the limitations identified in previous studies served as the foundation for our presented IPDS.

Janet et al. [25] used DL approaches to detect phishing. The classification methods, Support Vector Machine (SVM), ANN, and RF, were all used. They discovered that the RF algorithm performed well. Rishi Kotak [26] detected phishing attacks using various DL methods. Finally, Rabab et al. [27] tested various DL models for phishing detection and discovered that random forests performed better.

A table of past paper references is provided below as shown in Table 1, along with their methodology, classification method, performance evaluation, training, and testing accuracy.

The literature review of Phishing Detection in Modern Security with URLs has revealed several possible research gaps in this field. One such gap is the lack of standardization in identifying phishing URLs, which makes it difficult for researchers to compare results and organizations to implement effective phishing detection systems. Another gap is the limited research on real-world data, as most studies use synthetic or artificially generated datasets that may not accurately reflect the complexity of real-world phishing attacks. Additionally, many studies focus solely on the technical aspects of phishing detection, without considering how users interact with phishing URLs. Understanding user behavior and decision-making processes is critical for developing effective phishing detection strategies. Finally, there is a limited focus on emerging threats, with many studies concentrating on detecting known types of phishing attacks without considering new attack vectors. Our proposed model aims to fill these gaps by utilizing a CNN-based model for precise classification that distinguishes legitimate websites from phishing websites. We evaluate the performance of our model on the PhishTank dataset and show that it achieves high accuracy rates and outperforms previous state-of-the-art models. Moreover, we created a real dataset by crawling 10,000 phishing URLs from PhishTank and 10,000 legitimate websites and then ran experiments using standard evaluation metrics on the data sets. Our model is founded on integrated and DL and considers user behavior and emerging threats to provide a more effective phishing detection strategy.

## 3. Research Methodology

This section describes a method for identifying phishing websites using a combination of DL techniques and ML algorithms. The proposed approach involves four steps that work together to achieve high accuracy in identifying phishing websites. The first step involves the dataset and dataset preparation and preprocessing. The second step involves converting URL data into a character vector using a technique called character embedding. Character embedding is a method for representing text data in a numerical format that can be processed by DL algorithms. This allows the URL data to be processed and analyzed by the subsequent components of the approach. The third step involves using a CNN to analyze the character vectors obtained from the first component. CNNs are a type of DL algorithm commonly used for image analysis, but they can also be applied to other types of data, such as text. The modified URL data obtained from the character embedding approach are used to create and train a better CNN network, which is then able to distinguish phishing websites from legitimate websites with high accuracy. Finally, the fourth step involves retrieving the URL properties to obtain the characteristics of the different layers of the CNN network. This allows researchers to better understand the inner workings of the trained model and identify which features of the URL data are most important for identifying phishing websites.

### 3.1. Dataset

#### 3.1.1. Phish Tank Data Set

A data set of authenticated fake websites is kept up by the free, community-driven service called PhishTank. The data set is compiled from user reports of suspicious websites and verified by an expert team. The PhishTank data set contains information such as the phishing website’s URL, the date it was first reported, and whether it is still active. The data set is accessible through an API or by downloading a CSV file. It is intended to be used for anti-phishing research and development and training and testing of DL models. Because the data set is updated daily, it is critical to use the most recent data set.

#### 3.1.2. Data Description

Table 2 shows the data set, which contains seven columns containing the following information:Phish ID: This is the unique identifier that PhishTank uses to identify phishing submissions. This ID serves as a link for all data in PhishTank.URL Phishing specifics: You may view details about a phish, such as a screenshot and user comments, by visiting the PhishTank descriptive URL for the website.URL the fake URL: There may be a CDATA section in the XDL feeds, but this is always a string.Submission time: This is the time and date that PhishTank was notified about this phish. This date is formatted using ISO 8601.Verified: This indicates whether or not our community has validated this scam. Because these data sets include validated phishes, it contains a yes string.Verification time: This is the time and date that our community determined the scam to be legitimate. This date is formatted using ISO 8601.Online: This indicates whether the phish is active and accessible online. Because we only supply web-based phishes in these metadata, this string is always “yes.”Target: This is the name of the company or brand that the phish is spoofing if it is identified.

**Table 2 sensors-23-04403-t002:** Phishing Data Set.

Phish ID	URL	Phish Detail URL	Submission time	Verified	Verification time	Online	Target
8018078	https://ipchiro.com/assets/sella	http://www.phishtank.com/phish_detail.php?phis…	2023-01-27T06:57:11+00:00	yes	023-01-27T07:04:29+00:00	yes	other
8018074	http://area-clienti.bambinos.it/st	http://www.phishtank.com/phish_detail.php?phis…	2023-01-27T06:53:06+00:00	yes	2023-01-27T07:04:29+00:00	yes	other
8018075	https://ipchiro.com/assets/managehosting/	http://www.phishtank.com/phish_detail.php?phis…	2023-01-27T06:53:06+00:00	yes	2023-01-27T07:04:30+00:00	yes	other
8018073	http://area-clienti.bambinos.it/st/	http://www.phishtank.com/phish_detail.php?phis…	2023-01-27T06:53:04+00:00	yes	2023-01-27T07:04:30+00:00	yes	other
8018069	https://aibudgets.com/meta/	http://www.phishtank.com/phish_detail.php?phis…	2023-01-27T06:29:31+00:00	yes	2023-01-27T06:33:10+00:00	yes	other

The above detail shows the data set collected from the phish tank website and all of the relevant information. It is important to note that these data are updated regularly, and we used the most recent data set available while modeling. The next step is to collect the data set of legitimate websites and compare it with phishing websites while extracting features.

#### 3.1.3. Legitimate Data Set

The legitimate URLs are obtained from the open data sets of the University of New Brunswick and presented in the following Table 3.

### 3.2. Feature Extraction

Several of the following features can be extracted for phishing detection using the PhishTank data set:Domain features: Extracting information such as the age of the hostname, the number of domain names, and the presence of specific words or terms in the field may assist in detecting phishing sites.URL Features: Extracting information such as the width of the URL, the inclusion of specific characters or text in the URL, or the quantity of dots in the web address can help detect phishing sites.HTDL features: Extracting features from a website’s HTDL source code, such as the presence of specific keywords or phrases.Content features: Extracting features such as the presence of specific keywords or phrases in a website’s content, the text’s sentiment, and the use of specific language can help detect phishing sites.Link features: Extracting the number of links on a page and the number of external links and links to other phishing sites can help detect phishing sites.WHOIS features: Extracting features from WHOIS records, such as the age of the domain, registrar, and owner’s contact information, can help detect phishing sites.SSL/TLS certificates: Extracting features from SSL/TLS certificates, such as the validity period of the certificate, the issuer of the certificate, and the presence of certain characters or keywords in the certificate, can help detect phishing sites.

In this study, we extracted the following features:

#### 3.2.1. Domain

The domain of a URL identifies the website or webpage being accessed. It is part of the URL that comes after the “http://” or “https://” and before the first “/.”

The domain is usually used to identify the organization or individual that owns and operates the website and can be used to identify the country or top-level domain to which the website belongs.

In phishing detection, the domain of the URL is used as a feature for identifying phishing websites. For example, a phishing website may use a domain that closely resembles the domain of a legitimate website to trick users into visiting the phishing website.

#### 3.2.2. URL IP

IP addresses may be used in place of domain names in URLs. If an IP address is used instead of the domain name, we can be positive that someone is attempting to steal personal information using this URL. This feature has a value of 1 if the domain component of the URL includes an IP address and 0 elsewhere.

#### 3.2.3. URL Length

This determines the URL’s length. Scammers can use long URLs to conceal the dubious portion in the address bar. In this project, a URL is classed as phishing if it has more than 54 characters but is otherwise legal. The value assigned to this characteristic is 1 (phishing) or 0 if the URL length exceeds 54 (legitimate).

#### 3.2.4. URL Redirection

This checks the existence of the URL. The visitor will be sent to a different page if the URL path contains the character “/.” The value assigned to this attribute is 1 or 0 (else) if the “//” character appears anywhere in the URL other than after the protocol.

#### 3.2.5. Http Check

This makes sure the “http/https” characters are included in the host name component of the address. To trick consumers, phishers might add an “https” certificate to the site.

#### 3.2.6. Short URLs

By using this method, it is possible to significantly shorten a URL on the “www” while keeping linking to the desired home page. If the address uses URL shortening services, this feature is set to 1; otherwise, it is set to 0.

#### 3.2.7. Checking Prefix

Check to determine whether the domain of the URL contains the symbol “-.” The value allocated to this feature is 1. Otherwise, it is 0 if the URL comprises the sign “-” in the hostname component of the URL. These features are extracted for phishing and legitimate websites, assigned a 1 or 0 label, and concatenated in one data store.

### 3.3. Data Visualization

There are two labels for each of the given features: 1 if the website is a phishing website and 0 if the website is legitimate. In the feature extraction section, seven features were extracted due to the data size of 20,000: 10,000 phishing websites and 10,000 legitimate websites. A graphic depiction of all of the qualities would assist in identifying authentic and fraudulent websites.

Figure 2 depicts a heat map visualization of the features. A heat map is a data visualization tool for numerical and categorical data. It displays the data in a 2D table format with the values assigned together, indicating a significant variation in the data values.

The URL, redirection, https, TinyURL, prefix/suffix, and label are all displayed in the data visualization graph. This technique employs a variety of interactive visuals within a specific context to assist people in comprehending and making sense of large amounts of data. As shown in Figure 3, the above output shows the histogram for each attribute in the data set.

From the above visualization phishing websites have the following:A large URL lengthNo IPNo https domainNo redirectionFew prefixes and suffixesA shortened URL length

### 3.4. Data Preparation and Cleaning

The stored data are then divided into features and labels in the following step. The first seven columns are chosen for the features section, and the final label column is chosen as a response. Next, this data set is divided into three sections: training, testing, and validation, which will be used to train, validate, and predict the data set using a trained DL model. To clean the data, the data set is checked for any null entries. When it comes to data analysis, dealing with missing data or abnormal inputs is crucial, particularly in real production environments. To handle such issues, several methods are available, each with its own strengths and limitations. Imputation is a common method that involves estimating the missing values based on observed values in the dataset. There are several types of imputation, including mean imputation, regression imputation, and hot-deck imputation. Another method is removal, which involves removing records with missing or abnormal inputs from the dataset. This approach is only advisable if the missing values are insignificant or do not affect the analysis results. Interpolation, on the other hand, involves estimating the missing values based on the values before and after the missing value, using linear interpolation or spline interpolation. Data augmentation is another method that generates new data points to fill in missing values or abnormal inputs. This approach is common in machine learning applications. Lastly, outlier detection is a method that identifies and removes records with abnormal inputs from the dataset, using z-score, boxplot, or Local Outlier Factor (LOF). The selection of a particular method depends on the nature and extent of the missing data or abnormal inputs in the dataset, as well as the type of analysis being performed.

There are no null entries in the Phishtank dataset, so the data set, which has 20,000 rows and eight columns, is divided into 30% testing and 70% training. This split has been chosen based on the standard practice in the field of ML or based on previous research in the area.

### 3.5. The Proposed Method

Using CNN, character embedding, and RFs, this section describes how to identify phishing websites. Figure 4 shows the general layout of the proposed approach. The strategy for identifying phishing websites discussed in this study consists of three distinct components. URL data are first converted into a character vector using the character embedding approach. The same data format of the transformed and original URLs makes it possible to distinguish phishing websites from legitimate websites. The modified URL data are then used to create and train a better CNN network. Finally, after the model has been trained, the URL properties are retrieved to obtain the characteristics of the different layers of the CNN network.

#### 3.5.1. Model Development

The methodology for the 1D CNN model for phishing detection using the PhishTank data set is as follows:

##### Gathering of Data and Preprocessing

Data sets from the PhishTank website were gathered and downloaded.The data set contains a list of URLs reported as phishing sites.The data set was preprocessed by removing irrelevant columns, handling missing values, and splitting them into training and testing sets.

##### Data Representation

Because the data set contains URLs, they were represented as character-level embeddings.A 1D CNN model takes in 1D input, so character-level embeddings were chosen as the representation.The input shape was set as the shape of the first sample in the training set.Training split:

We chose a 70–30 ratio for training and testing, respectively. This means that 70% of the dataset was used for training the model, and the remaining 30% was used for evaluating the model’s performance. The reason for choosing this ratio is to ensure that the model is not overfitting to the training data. If too little data are used for training, the model may not be able to learn the underlying patterns in the data. On the other hand, if too much data are used for training, the model may memorize the training data instead of learning the underlying patterns. Therefore, we chose a ratio that would provide enough data for the model to learn the underlying patterns while also ensuring that the model does not overfit the training data. The PhishTank dataset consists of 20,000 rows and eight columns, which provided us with enough data to train and test our model.

#### 3.5.2. Model Architecture

The character embedding approach creates and trains an enhanced CNN network based on the altered URL matrix. The back-propagation approach continually updates several model parameters after the CNN network has been trained using the URL learning algorithm. After the model training completion, multilayer URL properties are recovered from the CNN network. The architecture of the CNN model is shown in Figure 5. It has seven layers, starting with the input layer and going to the seventh, which is a layer with pooling, convolutional, linear 1 and 2, and linear six layers as the output layers.

A sequential model was used, where several layers were added one after the other.A 1D convolutional layer with 128 filters, a two-kernel size, and a ReLU transfer function made up of the top layer (Conv1D).BatchNormalization and Dropout layers were added after the Conv1D layer to normalize the data and prevent overfitting.BatchNormalization and Dropout layers were added again after the second Conv1D layer.A flattened layer was added to flatten the output of the previous layers.A layer with 512 units and a linear transfer function was added to construct an ultimately linked layer.Dropout was added to prevent overfitting.For binary classification, the final layer was a dense layer with 1 unit and a ReLU activation function.

##### Training and Evaluation

The evaluation metrics used were precision, recall, accuracy, F1 score, and recall.More data and other hyperparameters might be employed to fine-tune the model and enhance performance. Finally, using the test set, the model’s effectiveness was evaluated.

In the case of the proposed method for identifying phishing websites, the number of filters used in the 1D convolutional layer was chosen to be 128. This number was likely selected based on a trade-off between the model’s ability to extract relevant features from the input data and computational efficiency. With 128 filters, the model can analyze the input data at a relatively high level of detail while still maintaining computational efficiency. The number of units within the layer is 512, which may have been chosen based on the complexity of the problem being solved. A larger number of units may allow the model to better capture complex relationships within the input data, but may also increase the risk of overfitting. The dropout rates of 0.2 and 0.5 are used in the model to prevent overfitting. Dropout is a regularization technique that randomly drops out (sets to zero) some of the neurons in the layer during training. This can help prevent overfitting by encouraging the model to learn more general features of the input data. The specific dropout rates of 0.2 and 0.5 were likely selected based on experimentation and tuning to achieve the best performance on the problem being solved.

##### D CNN Model for Classification

A 1D CNN model is a CNN model that only has one dimension, such as text or time series data. An input layer, one or more convolutional layers, pooling layers, and an output layer make up the fundamental building blocks of a 1D CNN model. A 1D CNN model’s input layer receives input data, typically preprocessed and transformed into numerical representations such as tokenized text or time series data. A 1D CNN model’s convolutional layers analyze the input data by applying filters to them. These filters move over the input data, extracting features and generating a new data representation known as a feature map. The size and number of filters may be adjusted to extract various kinds of characteristics from the input data. The pooling layers of a 1D CNN model are responsible for lowering the feature dimensionality. It is typically accomplished by performing a pooling operation on the feature maps, such as maximum or average pooling. Pooling layers reduce model complexity by making it less sensitive to minor variations in input data.

The output layer of a 1D CNN model produces the final prediction. It is usually accomplished by applying a fully connected layer to the feature maps and using it to categorize the input data into one or more groups. 1D CNNs can be trained using the SGD back-propagation algorithm or other optimization algorithms such as AdaGrad, Adam, and others.

In summary, a 1D CNN model is a form of NN designed to handle data with only one dimension, such as text or time series data. An input layer, one or more convolutional layers, one or more pooling layers, and an output layer are the components of a 1D CNN model, which are shown in Figure 6 as their fundamental structure. These layers work together to extract properties from incoming data and use them to make predictions.

### 3.6. Model Architecture

The architecture of the proposed model for PD is as follows: The CNN phishing detection model defines a 1D CNN classification model using the Keras library in Python, as depicted in Figure 7. The model is defined using the sequential function, which creates an empty model that can be built layer by layer.

The first layer creates a 1D convolutional layer with 128 filters of size 2, using the ReLU activation function and an input shape specified as X_train [0] shape. The input shape is specified, so the model knows how to expect the input data.The second layer includes a BN layer, normalizing the preceding layer’s output by substituting the batch mean and dividing it by the batch SD. It can aid in reducing overfitting and enhancing model performance.During training, 20% of the neurons from the preceding layer’s dropout layer, which is introduced in the third layer and has a dropout rate of 0.2, are randomly removed. It can also aid in reducing overfitting.After constructing a second 1D convolutional layer with 256 filters of size 2 and the ReLU activation function, the subsequent two layers—one more BN layer and one with a dropout rate of 0.5—are added. Finally, to be transmitted to a dense layer, the output of the convolutional layers is flattened in the sixth layer.The nonlinear activation function is handled by a dense layer with 512 neurons and a dropout rate of 0.5 in the seventh layer, which is also dense.The activated sigmoid function creates a probability output between 0 and 1, representing the likelihood that the input signal corresponds to a specific class.

Table 4 shows how the given model creates a 1D CNN classification model in Python using the Keras library. Layers in the model include 1D convolutional layers, batch normalization layers, dropout layers, and dense layers. These layers work together to extract features from the input data, classify it into one or more categories, and reduce overfitting. This model produces a probability between 0 and 1, which can be articulated as the possibility that the input data relate to one of several classes.

#### 3.6.1. Convolution 1D Layer

In an NN, a 1D convolutional layer uses a set of convolutional filters for the input data. Small filters are dropped over the input data to extract features. A feature map is the performance of a one-dimensional convolutional layer.

The kernel size, the number of filters, and transfer functions are the three main assumptions passed to the Conv1D layer. For example, the first Conv1D layer in the proposed model has 128 filters of size two and employs the ReLU activation function. Therefore, it implies that the layer will generate 128 output extracted features, each with the same number of variables as the data input. However, a width of 2 and the output will be processed by the activation function of ReLU before being passed to the next layer.

#### 3.6.2. Batch Normalization (BN) Layer

BN is a technique for normalizing the activations of a layer in a Neural Network. It is applied to a layer’s output before passing it to the next layer. BN aims to stabilize the training process and make the model less adaptive to the input data scale. This step standardizes the input data for the next layer and makes the model more resistant to changes in the input data.

BatchNormalization () is implemented after the initial and second Conv1D layers in the offered code. It normalizes the output of these layers before passing it to the next layer. For example, the BN layer operates by normalizing the input data with zero mean and unit variance. It improves training stability and speeds up the training process.

#### 3.6.3. Dropout Layer

Dropout is a batch normalization method used to reduce overfitting in NN. It works by randomly dropping out a certain proportion of the neurons during each training iteration. It helps prevent the network from relying too much on any single neuron and forces the network to learn multiple independent representations of the data.

The Dropout layer is applied after the first and second Conv1D layers, the first Dense layer, and the Conv1D layers. By doing this, the network is prevented from overfitting to the training set of data, and the model is strengthened against changes in the input data. The dropout rate of 0.2, 0.5 and 0.5 means that during each iteration, 20%, 50%, and 50% of the neurons in the first, second, and fourth layers, will be set to zero, respectively. It helps to prevent the network from relying too much on any single neuron.

#### 3.6.4. Flatten Layer

The Flatten layer is used to convert the multidimensional output of the previous layer to a one-dimensional array. It is required before passing the output to a fully connected layer (Dense layer) which expects a one-dimensional input.

The Flatten layer is applied after the second Conv1D layer in the provided code. It converts the multi-dimensional performance of the second Conv1D layer to a one-dimensional cluster, which is then carried to the next Dense layer.

The Flatten layer reshapes the performance of the second Conv1D layer into a one-dimensional array, passed to the Dense layer because the Dense layer anticipates a one-dimensional input, the second Conv1D layer’s multidimensional output should be reshaped to a one-dimensional array.

#### 3.6.5. Dense Layer

A dense layer, or convolution layer, is a type of layer in a neural network that connects all of the neural connections from the preceding layer to the neurons from the input layers. There are two dense layers in the presented model, including one with 512 neurons, whereas the other has one neuron. The first dense layer includes 512 neurons with an actual kernel function. The second dense level comprises one neuron with an activated sigmoid function.

Following the Flatten layer, the first Dense layer is applied. It takes the flattened layer’s one-dimensional array output and applies 512 neurons. Then, using the second dense layer, the network’s ultimate output is produced. It just contains one neuron and activates via the “sigmoid” method. With the help of the sigmoid function, a particular kind of activation, the input is transformed into a value between 0 and 1, which may indicate a particular class’s likelihood.

To learn the high-level characteristics from the output of the preceding layer, the first dense layer, which has 512 neurons, is employed. The model may learn more complicated data representations because of the rising prevalence of parameters used by the model. A single neuron from the second dense layer is employed to obtain the network’s final output. It is in charge of making the ultimate conclusion regarding the input based on the previously learned features.

### 3.7. Hyperparameters Details

Hyperparameters are parameters that are not learned during the training process of a neural network but rather set before training begins. They include the following:Batch size: Larger batch sizes result in faster training progress but necessitate more memory. Although smaller batch sizes necessitate less memory, training progress may be slower.Epoch: This is the number of times the model will traverse the entire training data set during training. The more epochs you run, the better the model gets, up to a point.Loss: PD and other binary classification problems frequently employ the binary cross-entropy gradient descent. The difference between the simple probability density function and what is anticipated is calculated.Optimizer: The optimizer is used to update the model’s parameters based on the gradients computed during back-propagation. Adam is a widely used optimizer that uses the gradient of the parameters and the moving average of the gradient to update the parameters.

All of these parameters are crucial in the classification model. The number of epochs and the batch size are used to train the model to determine how much computing is necessary. The model’s parameters are updated using the optimizer, and the model’s effectiveness is evaluated using the loss function. The categorization problem determines optimizer and loss functions. The Adam optimizer and binary cross-entropy loss are employed because of the binary classification nature of the problem. This means that the model should be capable of predicting two classes.

The number of epochs is determined by balancing computation time and model precision. The model may be able to learn more features with more epochs, but it will also take more computation time. Batch size is also essential in terms of computation time and memory usage. Larger batch sizes require more memory but result in faster progress in training, whereas smaller batch sizes require less memory, but training progress may be slower.

In summary, these hyperparameters control the model’s accuracy and computational time. The classification problem, the available computational resources, and the trade-off between computation time and accuracy are considered while choosing it. For epoch 50, the model is trained with a batch size of 32. The end-period training results are 98.97% correct, as shown in Table 5.

Number of 128 filters for the 1D convolutional layer: The choice of 128 filters is arbitrary and can vary depending on the specific problem and dataset. However, a larger number of filters can capture more complex patterns and features in the data, potentially leading to better performance. In general, it is recommended to start with a smaller number of filters and increase gradually to avoid overfitting. A number of 512 units within the layer: The choice of 512 units in the dense layer is again arbitrary and can vary depending on the complexity of the problem and dataset. A larger number of units can capture more complex relationships in the data, but can also lead to overfitting. It is generally recommended to start with a smaller number of units and increase gradually to find the optimal balance between model complexity and performance.

Dropout rates of 0.2 and 0.5: Dropout is a regularization technique that randomly drops out a certain percentage of neurons during training to prevent overfitting. The choice of dropout rates depends on the complexity of the model and dataset. A higher dropout rate can be useful for larger and more complex models, while a lower dropout rate may be sufficient for simpler models. The choice of specific dropout rates in your model is based on experimentation and tuning.

Other parameters: Other parameters such as activation function, kernel size, and batch normalization are also important in determining the performance of the model. The choice of specific parameters is based on experimentation and tuning and can vary depending on the specific problem and dataset. The choice of the specific model architecture and hyperparameters is based on experimentation and tuning to achieve the best performance on the specific problem and dataset at hand.

## 4. Results and Analysis

The method was programmed and tested on a machine with an Intel Core i7 processor and 16GB RAM, running Ubuntu 18.04 LTS operating system. We used Python programming language version 3.7.4 and TensorFlow framework version 2.0.0 for implementing the proposed method. Model evaluation is assessing the accuracy of a classification model on a data set. It aids in determining the model’s ability to predict the target classes accurately. The evaluation metrics used are determined by the kind of binary classification and the specific requirements of the problem. Following are the evaluation metrics for a binary classification phishing detection model:

Based on the results of the research conducted within the manuscript several future directions involved parties should take in profit from the findings of the study. While the model achieved high accuracy, it is important to note that it was trained and tested on a specific dataset, PhishTankit is important to evaluate the model’s performance on other datasets to validate its effectiveness in a wider range of scenarios. This would involve collecting and labeling additional datasets and testing the model’s performance on these new datasets. Phishing attackers are constantly evolving their techniques, so it is important to keep updating the model to keep up with these changes. One way to achieve this is to continuously monitor the performance of the model and retrain it periodically with new datasets. The model can also be updated to incorporate additional features that are relevant to phishing attacks, such as the use of social engineering tactics. The model can be integrated into existing security systems to enhance their effectiveness in detecting and preventing phishing attacks. This would involve developing an API or a plugin that can be easily integrated into various security systems. Additionally, the model can be used to provide real-time feedback to users, alerting them of potential phishing attacks and advising them on how to avoid falling victim to these attacks. Fourth, it is important to consider the ethical implications of using the model. The model is trained on user data, and as such, there are privacy concerns that must be addressed. The involved parties should take steps to ensure that the model is used in a way that respects users’ privacy rights and complies with relevant data protection regulations. The findings of the study provide a promising approach to detecting and preventing phishing attacks using deep learning. By continuing to develop and improve the model, and by integrating it into existing security systems, the involved parties can enhance their ability to protect users from phishing attacks. However, it is important to remain vigilant and continue to monitor the model’s performance and address any ethical concerns that arise. Future research in the field of phishing detection could focus on further improving the model design. One potential area for improvement is the development of more advanced pre-processing techniques to enhance the quality of the data fed into the model. Alternative deep learning architectures, such as recurrent neural networks (RNNs) or transformers, could be explored to compare their performance against the 1D CNN model used in this study. In terms of data, expanding the data set beyond PhishTank could provide a more comprehensive evaluation of the model’s effectiveness. Including additional types of phishing attacks, such as spear phishing or pharming, could also help to further validate the model’s ability to detect a wider range of phishing attempts. While the study achieved high accuracy, there is still room for improvement in terms of reducing false positives and false negatives. Investigating additional features that could be extracted from URLs, such as the content of the website itself, could potentially enhance the model’s ability to distinguish between legitimate and phishing URLs, it may be useful to evaluate the model’s performance against more advanced phishing attacks that utilize more sophisticated techniques, such as social engineering or machine learning-based approaches. By doing so, the effectiveness of the model could be further validated and potential areas for improvement could be identified. There are several areas for future research in the field of phishing detection, including improving the model design, expanding the data set, reducing false positives and false negatives, and evaluating the model against more advanced phishing attacks. By addressing these areas, researchers can continue to enhance the effectiveness of phishing detection techniques and better protect individuals and organizations from these malicious attacks.

1 Score: Recall and precision are achieved through arithmetic.AUC-ROC curve: This region is underneath the ROC curve.Confusion matrix: This gives an overview of True Negative, True Positive, False Negative, and False Positive predictions.Precision: TP forecasts as a percentage of all optimistic predictions.Accuracy: The percentage of samples that were correctly categorized.Recall (Sensitivity or TPR): The proportion of accurately identified samples as positive.

As shown by the training accuracy of 98.77% and validation accuracy of 98.01% in the classification model for phishing detection, there is neither overfitting nor underfitting in the model. It indicates that the model performs well on training data.

Training accuracy measures how well a classification model predicts the target classes on the training data set. It indicates how well the model learned the patterns in the training data and can be used to monitor overfitting or underfitting, as shown in Figure 8.

High training accuracy is desirable because it indicates that the model can correctly classify the training data. However, assessing the model’s performance on a validation or test data set is also critical because high training accuracy only sometimes implies good performance on unseen data. Overfitting occurs when a model becomes too complex and memorizes the training data, resulting in high training accuracy but poor performance on unseen data.

The above two graphs, shown in Figure 8 and Figure 9, show that the model has a high training accuracy and a validation accuracy close to it, indicating that the model performs well on unseen data. The difference between the predicted outputs of a classification model and the actual target values for a given training data set is measured as training loss. It is used during training as an optimization objective to minimize training loss so the model’s predictions are as close to the true targets as possible. A lower validation loss signifies that the model better fits the learning algorithm. In contrast, a high learning loss indicates that the model is experiencing problems learning the patterns in the data. Monitoring the training loss during training can assist in detecting overfitting, where the model has become too complex and is memorizing the training data, and underfitting, where the model is too simple and unable to capture the patterns in the data.

### 4.1. Training Time Complexity

The computational complexity of training a DL model is influenced by various factors, such as the model’s complexity, the training data’s size, the optimization algorithm used, and the hardware used for training. In general, training a model with a more significant number of parameters or a more complex architecture will take longer to train. In addition, the optimization algorithm can also significantly impact the training time because some algorithms are more computationally expensive than others. Finally, training on a high-performance GPU can be much faster than training on a CPU. Training time complexity is usually expressed as an average-case or worst-case scenario, and actual training time can vary widely based on the specifics of the problem and implementation.

### 4.2. Confusion Matrix

Figure 10 depicts a confusion matrix, which is a table used to evaluate the accuracy of a classification model. It articulates how many FP, TP, FN, and TN predictions the model made. The following values can compute performance metrics, including recall, precision, accuracy, and F1 score:

The number of negative samples the model correctly identified as unfavorable is called TN.FN is the proportion of positive samples the model mistook for negative ones.TP measures how many positive samples the model identified adequately as positive.FP measures how many negative samples the model mistook for positive ones.

Figure 11 illustrates the suggested confusion matrix paradigm. The confusion matrix here offers a method for visualizing and evaluating the model’s performance and can aid in pinpointing areas that need improvement. For example, a high number of false negatives might indicate a need for the model to improve its ability to detect positive samples.

### 4.3. Evaluation Matrix

For problems involving binary classification where the target can be one of two classes (e.g., positive or negative), the metrics shown in Table 6 are used to evaluate the proposed model.

Table 7 shows the accuracy and loss performance model. It covers the evaluation metric values such as training, validation accuracy, and training and validation loss.

It is critical to remember that validation loss is only a measure of performance on the training data set and does not always reflect the model’s ability to generalize to new data. Evaluating the model on a validation or test data set provides a more comprehensive assessment of its performance. The above table shows that the model performs well on training and testing data with higher accuracy and low loss values.

### 4.4. ROC Curve

The ROC curve represents a binary classifier’s performance as the discrimination cut-off point is diversified. The regression model analyzes the FPR to the TPR at varied threshold values.

Figure 12 shows how the ROC curve can be used to visualize and evaluate the performance of various classifiers. A perfect classifier would possess a TPR of 1 and an FPR of 0, resulting in a moment in the ROC space’s upper-left corner. A random classifier would produce a diagonal line in the ROC curve varying from (0,0) to (1,1). The classifier better than random will result in a curve above the diagonal.

The AUC-ROC is a scalar summary of the ROC curve that evaluates the classifier’s ability to differentiate between negative and positive classes. For example, a classifier with an AUC-ROC of 1 is a perfect classifier, whereas a classifier with an AUC-ROC of 0.5 is no higher than random.

In our model, ROC shows that the model has perfect classification over seen and unseen data, with all other parameters showing the same performance.

Model retaining: When new data are encountered, they can be stored in a separate dataset or added to the existing dataset used for training the neural network. The new dataset can then be used to retrain the network using the same architecture and hyperparameters as before. Alternatively, if the new data are found to have different characteristics than the existing data, a new neural network model may need to be developed to handle the new data. The updated network can then be used to make predictions on the new data, and the results can be evaluated to determine if any further adjustments need to be made to the model. In general, the process of updating a neural network model with new data can be an iterative process that requires continuous evaluation and refinement to ensure optimal performance.

### 4.5. Comparative Analysis

Table 8 presents the results of various approaches that have been used to identify phishing websites, along with their corresponding accuracy rates and the dataset used to test their performance. The approaches include KNN, SVM, Random Forest Classifier, RNN, NLP, and our proposed approach, 1D CNN. It can be observed that our proposed approach using 1D CNN achieved the highest accuracy rate of 98.77% on the PhishTank dataset, outperforming all the other approaches listed in the table. This suggests that our approach is more effective in distinguishing phishing websites from legitimate ones compared to the other methods. The results also show that NLP, another DL-based approach, achieved an accuracy rate of 97.98%, which is slightly lower than our approach. The RNN-based approach also achieved a high accuracy rate of 97.4%. Meanwhile, the SVM and Random Forest Classifier approaches achieved accuracy rates of 94.13% and 94.26%, respectively. The KNN approach achieved the lowest accuracy rate among the listed methods at 87.98%.

## 5. Discussion

Phishing attacks remain among the most serious threats that must be effectively managed. DL algorithm advancements have recently led to the creation of effective phishing detection models. Although various variants have been developed so far, there continue to be obstacles to adaptable solutions. The proposed model aims to fill the current research gaps in phishing detection by utilizing a CNN-based model that can accurately classify legitimate websites and phishing websites. This approach is evaluated on the widely used PhishTank dataset, which consists of URL features for known phishing websites. The model achieves high accuracy rates and outperforms previous state-of-the-art models in terms of its false-positive rate. To create a more realistic dataset, the researchers crawled 10,000 phishing URLs from PhishTank and 10,000 legitimate websites to run experiments with standard evaluation metrics on these data sets. This model is founded on integration and DL, which allows for the consideration of user behavior and emerging threats in phishing attacks. The researchers acknowledge that user behavior is critical to developing effective phishing detection strategies, and their model takes this into account. Additionally, the model considers emerging threats and new attack vectors, which is an important aspect that many previous studies overlook. The process of detecting phishing websites is outlined, and the created data set is extensively examined. The outcomes of the studies demonstrate that the technique is reliable and has a low FPR.

Additionally, three main components of the method for detecting phishing websites are suggested in this paper. The character convolutional technique converts URL data into a particular set first. The converted URL data are then used to develop and train an improved CNN network. Figure 4 depicts the proposed method’s overall structure. The URL learning algorithm creates the CNN network, and the back-propagation technique continuously updates several of the model’s parameters. In this study, the CNN model’s architecture is shown in Figure 5. The input layer is the first of seven layers that make up CNN, whereas the output layer is the seventh. The output layers are linear 1 and 2, convolutional, and pooling.

In contrast, a 1D CNN model is a CNN designed to process data with a single dimension, such as text or time series data. An input layer, one or more entangled neighboring layers, and an output layer make up the fundamental components of a 1D-CNN model. Using filters, the convolutional layers of a 1D CNN model are responsible for analyzing the incoming data. These filters are slid over the data, resulting in an entirely new metadata known as a convolution layer. The number and size of filters can be changed to extract various types of characteristics from the data. The max-pooling of a 1D CNN model is responsible for lowering the dimension of the feature maps. This is typically accomplished by employing a pooling operation, such as maximum or average pooling. Finally, the final prediction is generated by the output layer.

The training accuracy is measured in the results section for a classification model that predicts the target classes on the training data set. A high accuracy rate demonstrates that the model characterizes the training data correctly. The ROC curve is a graphic depiction of the performance of a classification algorithm as the discrimination cut-off is changed. A perfect classifier has a TPR of 1 and an FPR of 0, resulting in a point in the top corner of the ROC space. Our model’s ROC is one, implying that it has perfect classification across both seen and unseen data, and all other parameters perform similarly. The ROC area is a scalar summary of the ROC curve that measures the classifier’s ability to distinguish between positive and negative classes.

Furthermore, some recent work on phishing detection has been conducted using the same PhishTank data set, as shown in Table 8. The table includes six different approaches, including KNN [37], SVM [40], random forest classifier [39], RNN [1], NLP [38], and the proposed model, which is a 1D CNN. The accuracy values for each approach are shown alongside the corresponding dataset used for training and testing. The KNN approach [37] achieved an accuracy of 87%, while the SVM [40] and random forest classifier [39] approaches both achieved a similar accuracy of around 94%. The RNN [1] and NLP [38] approaches performed better, with accuracies of 97.4% and 97.98%, respectively. Finally, the proposed 1D CNN model achieved the highest accuracy of 98.77%. Overall, these results demonstrate the effectiveness of using ML approaches for phishing detection, with the 1D CNN model proposed in this paper achieving the highest accuracy. Our study used a deep-learning 1D-CNN model with three critical layers: 1D convolution, batch normalization, and layer. CNN is a DL model that performs well in terms of accuracy. However, our model performed better than previous work because we used more layers and larger training sizes and extracted more features from the PhishTank data set, resulting in higher model accuracy. Finally, our model has 98.77% accuracy.

While the proposed approach for detecting phishing websites using a CNN-based model has shown promising results, there are some potential limitations to consider. Firstly, the approach has only been evaluated on the PhishTank dataset, which is widely used but may not be representative of all types of phishing attacks. This dataset bias could limit the generalizability of the proposed approach. Additionally, the proposed approach requires crawling and analyzing URLs, which may not be suitable for the real-time detection of phishing attacks. This could limit the applicability of the proposed approach in certain situations where real-time detection is critical. It is important to acknowledge these potential limitations and consider further research to address them.

The proposed method for detecting phishing sites using DL has the potential to be implemented in real-world environments to improve cybersecurity. The method employs a CNN for high-accuracy classification of URLs to distinguish between genuine and phishing sites. On the user’s side, our method could be implemented as a browser extension or plugin that intercepts URLs accessed by the user and performs phishing detection in real-time. The user would then be notified if a detected phishing attempt is made. Alternatively, the method could be implemented on the server or provider side, where URLs are checked before being delivered to users. This could be conducted using various mechanisms such as API calls or integrating the method into existing security systems. It is also possible to implement the method on both the user and server sides with specific adjustments, depending on the specific security needs and constraints of the environment. For example, in a large organization, implementing the method at the server side could provide an additional layer of protection, while also implementing it at the user side could provide additional protection for remote workers who may not be connected to the organization’s network. The developed method can be used by service providers to enhance the security of their systems and protect their users from phishing attacks. The method can be integrated into existing security systems as an additional layer of protection. Service providers can use the method to scan and filter out potentially malicious URLs before they reach their users, thereby reducing the risk of phishing attacks.

To improve cybersecurity and reduce the risk of falling victim to phishing attacks, organizations and individuals should consider implementing our proposed approach as part of their defense mechanism. Our approach achieved a high accuracy rate, making it an effective tool for detecting and preventing phishing attacks. Additionally, future research should focus on developing real-time detection methods that do not require crawling and analyzing URLs. This will enable the proposed approach to be more widely applicable in critical industries such as finance and healthcare. Finally, to improve the generalizability of phishing detection approaches, the dataset used for evaluation should be diversified to include a wider range of phishing attacks. Currently, the PhishTank dataset used in our study only includes attacks based solely on URL features. By reducing bias and improving the generalizability of the proposed approach, more effective and robust phishing detection methods can be developed.

## 6. Conclusions

This refinement, by using DL for detecting phishing attacks, has the potential to significantly reduce the number of phishing attacks and encourage caution in the future. Because the calculation is used in this case, a well-prepared model will identify URL vulnerability with fewer errors. The study’s findings demonstrated that DL algorithms could recognize and analyze URLs. Our initial impression is that the model created using the suggested method is more precise than the earlier ones.

This research aims to use a CNN algorithm to identify phishing using the PhishTank data set. The purpose was to create a conceptual framework that could identify phishing websites accurately and develop the current PD iteration. The CNN model is trained and assessed based on performance criteria, such as precision, accuracy, recall, and F1 -score, following preprocessing of the PhishTank data set. The area under the ROC curve, calculated as a scalar summary of the model’s capacity to distinguish between non-phishing and phishing websites, was constructed to highlight the model’s performance. In terms of performance on the PhishTank data set, the CNN model fared better than the existing models. It demonstrates the utility of using DL models for phishing detection and implies that the model can enhance the current state in this field. The research confirms the classification of phishing attacks by 1D-CNN and its preventative measures and consequences, with each algorithm having an accuracy of up to 98.77%. The overall performance score is used to evaluate the forecast and aid in future development.

In conclusion, the study’s findings demonstrate the efficacy of using a CNN model for PD and indicate a path forward for future studies in this area. Furthermore, the results show that this framework has the potential to dramatically improve PD’s efficacy and accuracy while giving users a superb alternative to the internet.

## Figures and Tables

**Figure 1 sensors-23-04403-f001:**
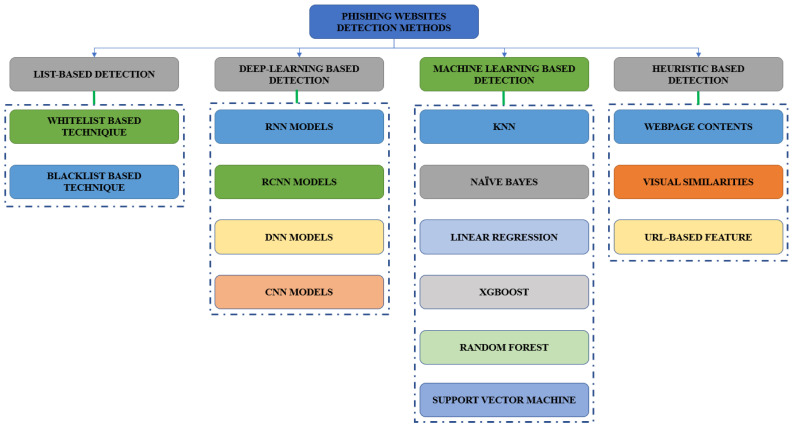
Various DL and ML Methods for Detecting Phishing.

**Figure 2 sensors-23-04403-f002:**
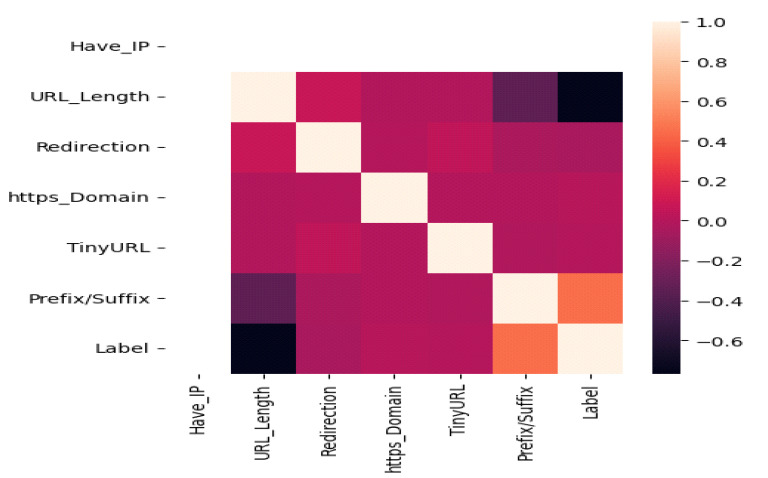
Heat Map Visualization of the Features.

**Figure 3 sensors-23-04403-f003:**
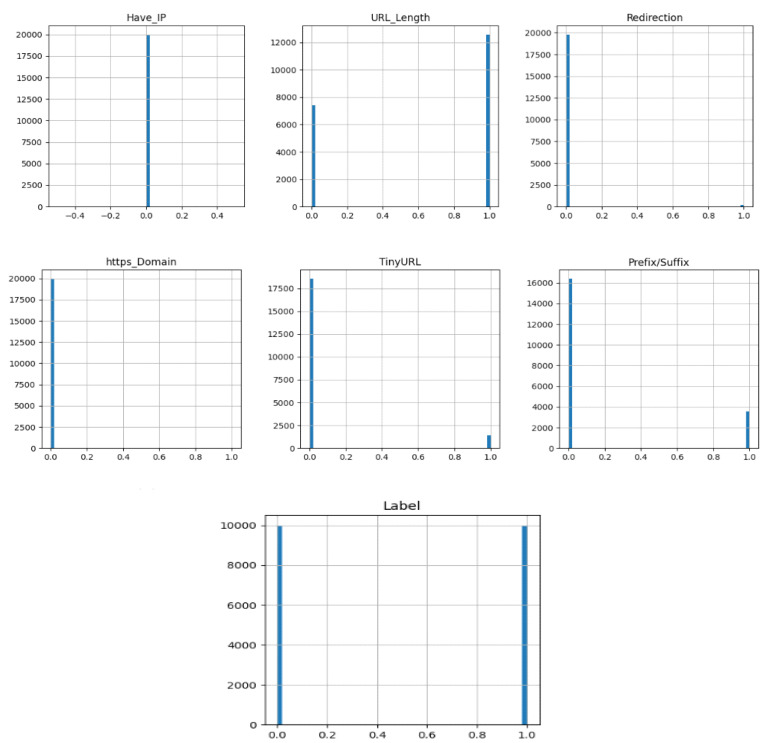
Features Visualization.

**Figure 4 sensors-23-04403-f004:**
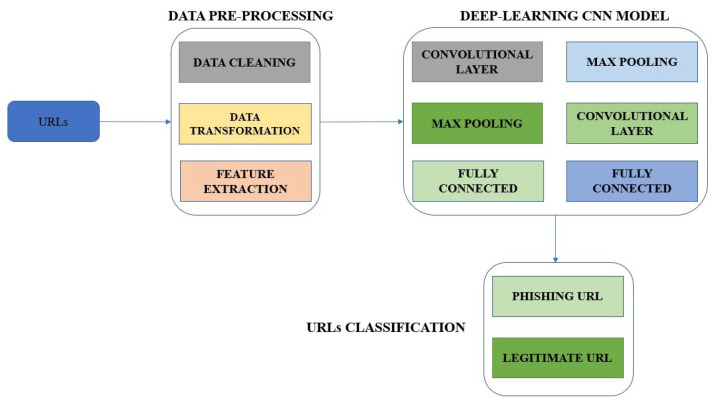
Framework of Proposed System with CNN Architecture.

**Figure 5 sensors-23-04403-f005:**
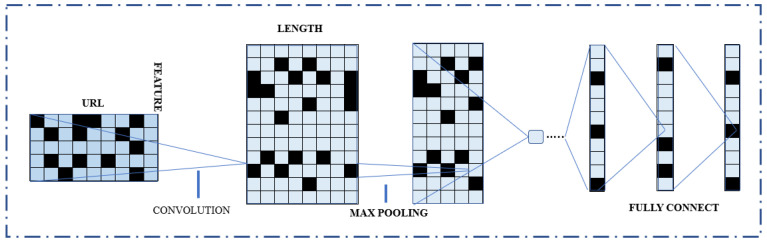
CNN Model Architecture.

**Figure 6 sensors-23-04403-f006:**
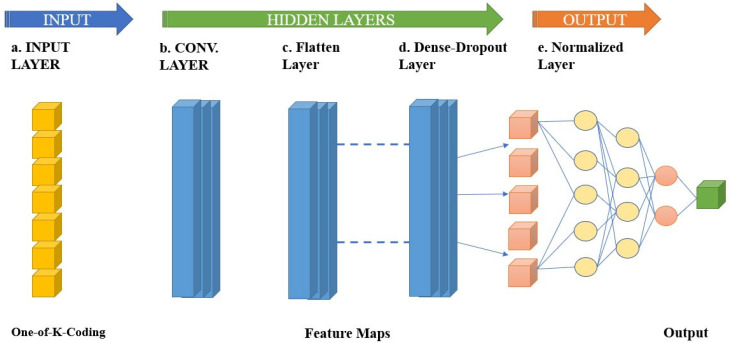
1D CNN Architecture for Classification.

**Figure 7 sensors-23-04403-f007:**
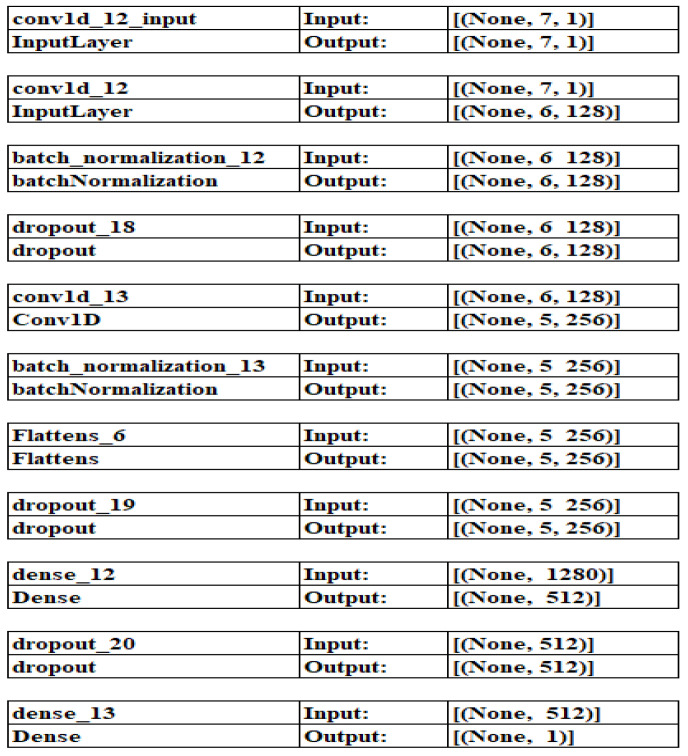
CNN architecture of the proposed model.

**Figure 8 sensors-23-04403-f008:**
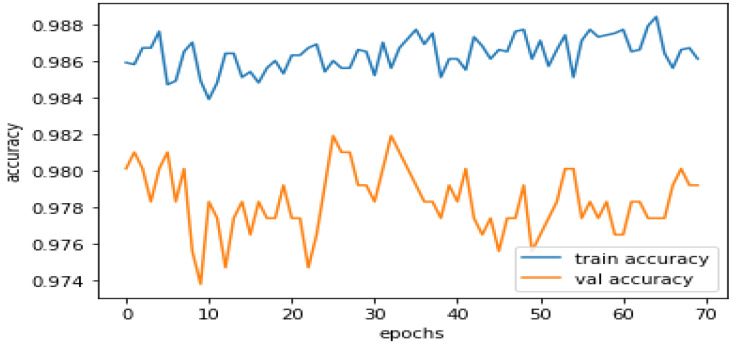
Training Accuracy Performance.

**Figure 9 sensors-23-04403-f009:**
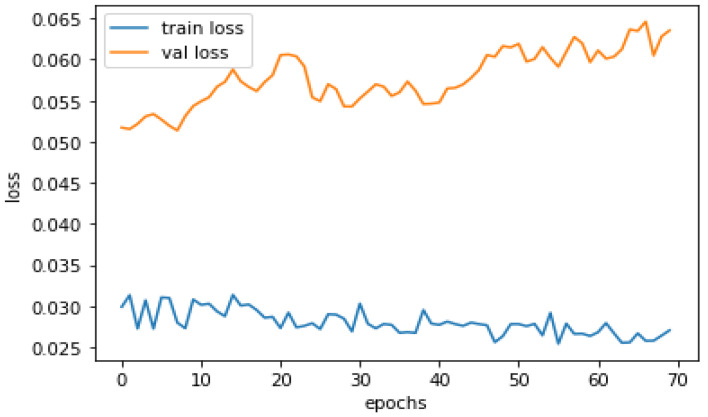
Training/Validation Loss Performance.

**Figure 10 sensors-23-04403-f010:**
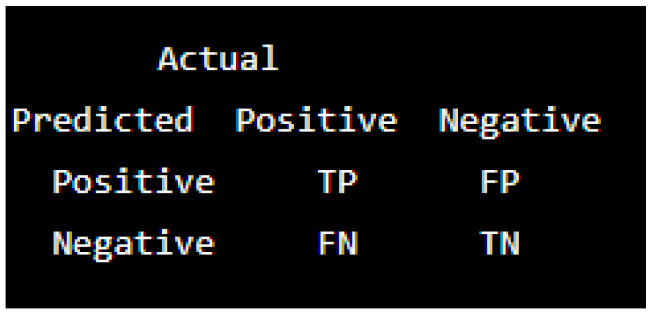
Representation of a Binary Confusion Matrix.

**Figure 11 sensors-23-04403-f011:**
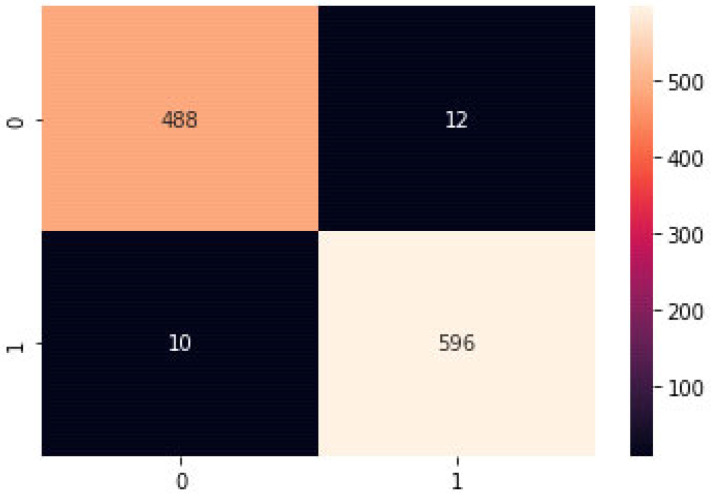
Proposed Model Confusion Matrix.

**Figure 12 sensors-23-04403-f012:**
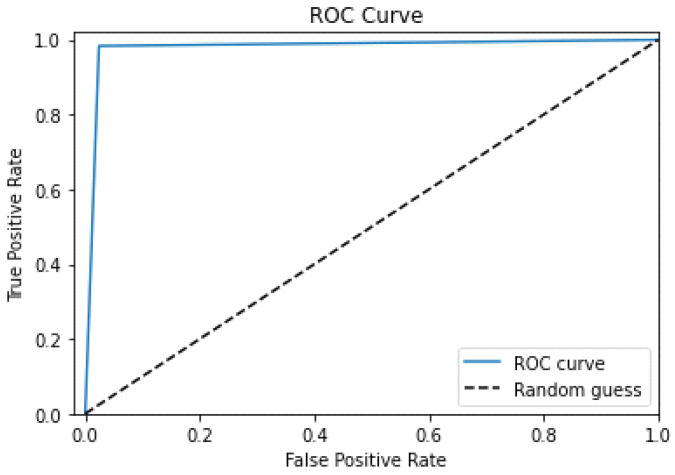
The curve for ROC.

**Table 1 sensors-23-04403-t001:** Past Paper References with Methodology, Data Set, Classification Method, Performance Evaluation, and Training and Testing Accuracy.

Ref.	Methodology	Data Set/ Parameters	Classification Method	Performance Evaluation	Training and Testing Accuracy
[10]	End-to-end automatic phishing webpage classification using DNN	Using a web crawler to read HTDL content	CNN and LSTM	Receiver Operating Characteristics, Precision, F1 score, Accuracy, Confusion Matrix,	2900 reputable URLs and 2300 extracted features
[11]	CNN, Deep Neural Network (DNN), DL, Machine-Learning	URLs at risk	Machine- Learning Based	Precision and run-time effectiveness	14% for testing and 86% for training accuracy
[12]	LSTM, CNN, DNN	Datasheet UCI	CNN and LSTM	Precision, F1 score, accuracy, confusion matrix	20% for testing and 80% for training accuracy
[13]	A quick solution approach based on DL	PhishTank, Alexa	RF, XGBoost, NB, Logistic Regression, DNN	Area Under the Curve (AUC), Precision, F1-Score, Accuracy	50% for testing and 50% for training accuracy
[15]	DL and CNN	Datasheet UCI	Machine- Learning Based CNN	Accuracy	10% for testing and 90% for training accuracy
[19]	LSTM and CNN	PhishTank data set	CNN and LSTM	Accuracy and classifier prediction	30% for testing and 70% for training accuracy
[25]	RNN, CNN, LSTM	PhishTank	The loss function for cross-entropy in LSTM	Precision and accuracy	20% for testing and 80% for training accuracy
[37]	CNN, DNN, bi-LSTM, ML	PhishTrim	Convolutional Structure	Accuracy	10% for testing and 90% for training accuracy

**Table 3 sensors-23-04403-t003:** Legitimate Websites.

Index	URLs
1	http://1337x.to/torrent/1110018/Blackhat-2015-..
2	http://1337x.to/torrent/1122940/Blackhat-2015-…
3	http://1337x.to/torrent/1124395/Fast-and-Furio…
4	http://1337x.to/torrent/1145504/Avengers-Age-o…
5	http://1337x.to/torrent/1160078/Avengers-age-o…

**Table 4 sensors-23-04403-t004:** CNN architecture properties.

Layer	Output Shape	Parameters
Input layer Conv1D	(None,6,128)	384
Batch normalization	(None,6,128)	512
Dropout	(None,6,128)	65,792
Conv1D	(None,5,128)	1024
Batch normalization	(None,5,128)	0
Drop out layer	(None,5,128)	0
Flatten	(None,1280)	0
Dense	(None,512)	655,872
Dropout	(None,512)	0
Output layer Dense	(None,1)	513

**Table 5 sensors-23-04403-t005:** Hyperparameters of the Proposed Model.

Hyperparameters	Properties
Epochs	50
Batch size	32
Optimizer	Adam
Loss	Binary cross entropy

**Table 6 sensors-23-04403-t006:** Evaluation Metrics of the Proposed Model.

Evaluation Metric	Performance Value
Accuracy score	0.981
Macro averaged precision	0.980
Micro averaged precision	0.9801
Macro averaged recall	0.979
Micro averaged recall	0.9801
Macro averaged F1 score	0.979
Micro averaged F1 score	0.981

**Table 7 sensors-23-04403-t007:** Accuracy and Loss Performance.

Evaluation Metric	Performance Value
Training accuracy	0.9877
Validation accuracy	0.9801
Training loss	0.0318
Validation loss	0.0537

**Table 8 sensors-23-04403-t008:** Related Work on Phishing Detection.

Ref.	Approach	Accuracy	Data Set
[1]	RNN	97.4%	PhishTank
[37]	KNN	87.98%	PhishTank
[38]	NLP	97.98%	PhishTank
[39]	Random Forest Classifier	94.26%	PhishTank
[40]	SVM	94.13%	PhishTank
Our Model	1D CNN	98.77%	PhishTank

## Data Availability

The dataset used in this study is freely available on request from https://www.phishtank.com/developer_info.php (accessed on 5 February 2023).

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
