# Peer review of "A Deep Learning-Based Innovative Technique for Phishing Detection in Modern Security with Uniform Resource Locators"

_sensors, 2023, doi:10.3390/s23094403_

Round 1
Reviewer 1 Report
This paper proposed a multi-layer deep learning technique for phishing websites detection using only URL features. The cleaning, training, development, and experiment of the deep learning technique are based on the Phish Tank data set, which makes sense. I think that this work is solid and the evaluation results are also good.
The organization/structure of this paper is clean/clear and the English is easy to be understood by readers.
My only concern here is as follows. This paper has 24-page in length. Is this too long for the Sensors journal or not? Is there a page limit for the paper publications for Sensors journal?
Author Response
Response Sheet
A Deep Learning-Based Innovative Technique for Phishing Detection in Modern Security with URLs
Reviewer 1
This paper proposed a multi-layer deep learning technique for phishing websites detection using only URL features. The cleaning, training, development, and experiment of the deep learning technique are based on the Phish Tank data set, which makes sense. I think that this work is solid and the evaluation results are also good.
The organization/structure of this paper is clean/clear and the English is easy to be understood by readers.
My only concern here is as follows. This paper has 24-page in length. Is this too long for the Sensors journal or not? Is there a page limit for the paper publications for Sensors journal?
Author Response: Thank you for your feedback. We appreciate your concern regarding the length of the paper. However, according to the submission guidelines provided by the Sensors journal, there is no specific page limit for paper publications. We have made every effort to ensure that the paper is concise, well-organized, and provides relevant information to the readers.

Reviewer 2 Report
Authors employ CNN for high-accuracy classification to distinguish genuine sites from phishing 22 sites, trained and tested the model on a PhishTank data set. Results show outperforms existing work. In binary-categorical loss and the Adam opti-29 mizer are used, the accuracy of the KNN, NLP, and RF models is 87%, 97.98%, and 94.26%. The proposed model is 98.77% accurate.
Modification:
1) There is a need to overall improve the language writing quality.
Additional comments:
1. What is the main question addressed by the research?
Deep learning is used to propose a model for phishing detection and the model is trained and tested on a dataset.2. Do you consider the topic original or relevant in the field? Does it
address a specific gap in the field? Yes, the work is original and fills a research gap.
3. What does it add to the subject area compared with other published
material? The proposed model improves the accuracy of phishing detection, which is better than previous work.
4. What specific improvements should the authors consider regarding the
methodology? What further controls should be considered? The authors already did enough in results and analysis.
5. Are the conclusions consistent with the evidence and arguments presented
and do they address the main question posed? Yes, the conclusion and work address the research question.
6. Are the references appropriate? yes, references are appropriate.
7. Please include any additional comments on the tables and figures.
Author Response
Response Sheet
A Deep Learning-Based Innovative Technique for Phishing Detection in Modern Security with URLs
Reviewer 2
Authors employ CNN for high-accuracy classification to distinguish genuine sites from phishing 22 sites, trained and tested the model on a PhishTank data set. Results show outperforms existing work. In binary-categorical loss and the Adam opti-29 mizer are used, the accuracy of the KNN, NLP, and RF models is 87%, 97.98%, and 94.26%. The proposed model is 98.77% accurate.
Modification:
1) There is a need to overall improve the language writing quality.
Author Response: We appreciate the reviewer’s feedback. We have improved the English of the paper as well as eliminated all the errors including grammatical errors, agreements mistakes, and punctuation errors.
Additional comments:
- What is the main question addressed by the research?
Deep learning is used to propose a model for phishing detection and the model is trained and tested on a dataset.
2. Do you consider the topic original or relevant in the field? Does it
address a specific gap in the field? Yes, the work is original and fills a research gap.
3. What does it add to the subject area compared with other published
material? The proposed model improves the accuracy of phishing detection, which is better than previous work.
4. What specific improvements should the authors consider regarding the
methodology? What further controls should be considered? The authors already did enough in results and analysis.
5. Are the conclusions consistent with the evidence and arguments presented
and do they address the main question posed? Yes, the conclusion and work address the research question.
6. Are the references appropriate? yes, references are appropriate.
Author Response: Thank you for positive response on our manuscript. We appreciate the reviewer’s feedback.

Reviewer 3 Report
I have reviewed the article "A Deep Learning-Based Innovative Technique for Phishing Detection in Modern Security with URLs", Manuscript ID: sensors-2294437 and I have identified a series of aspects that in my opinion must be addressed in order to bring a benefit to the manuscript.
In this paper, the authors present a deep learning-based approach for detecting phishing sites with a high level of accuracy. The manuscript under review is interesting. However, I consider that the article will benefit if the authors take into account the following remarks and address within the manuscript the signaled issues:
Remark 1: the main strong point of the manuscript consists in the fact that it approaches a very interesting topic for the experts in the field.
Remark 2: the main weak point of the manuscript under review consists in the lack of very important details regarding the implementation of the devised approach for phishing sites detection in a real-world working environment. Consequently, the manuscript will benefit if the authors make remarks and provide insights within the article as to how can the developed method be implemented in a real-world day to day working environment. Can the method be implemented at the user's side, the server's/providers of services side or it can be implemented with specific adjustments at both sides?
Remark 3: issues regarding the citations. Even if the authors have documented their research, there are some issues regarding the citations, the references. Therefore, in the actual form of the manuscript, several paragraphs from the Manuscript ID: sensors-2294437 have been re-written from another paper that is not even cited in the manuscript under review, namely "Dutta AK, Detecting phishing websites using machine learning technique. PLOS ONE 16(10): e0258361. https://doi.org/10.1371/journal.pone.0258361".
For exemplification, I will present only a few non-limitative examples, since the manuscript abounds in such situations where the paragraphs are properly re-written, but they are not cited to their original source:
At Lines 38-39, the authors state: "Phishing is a type of fraud involving technological and social approaches to collect financial and personal data from clients." This sentence can be found in an almost identical form in the above-mentioned paper of Dutta AK, as follows: "Phishing is a fraudulent technique that uses social and technological tricks to steal customer identification and financial credentials.".
At Lines 59-61, the authors state: "To obtain confidential information, criminals create illegal reproductions of legitimate websites and emails, typically from financial institutions or other organizations that deal with financial data. This email contains logos and slogans from a reputable company." This sentence can be found in an almost identical form in the above-mentioned paper of Dutta AK, as follows: "In order to receive confidential data, criminals develop unauthorized replicas of a real website and email, typically from a financial institution or other organization dealing with financial data. This e-mail is rendered using a legitimate company’s logos and slogans."
At Lines 64-65, the authors state: "…there is a significant risk of user data being misused. For these reasons, phishing is highly urgent, complex, and critical in today’s society". This sentence can be found in an almost identical form in the above-mentioned paper of Dutta AK, as follows: "There is a significant chance of exploitation of user information. For these reasons, phishing in modern society is highly urgent, challenging, and overly critical."
At Lines 67-69, the authors state: "If malicious code is installed on the website, installed malware and hackers may steal user information posing a significant threat to cybersecurity and user privacy.". This sentence can be found in an almost identical form in the above-mentioned paper of Dutta AK, as follows: " In the event of malicious code being implanted on the website, hackers may steal user information and install malware, which poses a serious risk to cybersecurity and user privacy."
The list of such situations can go on, but the previously presented examples are enough to be able to conclude that the authors' proposed study has strong similarities and therefore must be compared with the previous study. I consider that the manuscript under review will benefit a lot if the authors highlight clearly in the discussion section of their manuscript what are the main differences between their proposed problem and solution from the manuscript and the ones presented in the above-mentioned article. I consider a major omission not to cite the above-mentioned paper and I would like the authors of the Manuscript ID: sensors-2294437 to explain the reason why they have used excerpts of text from the previous paper, without even citing it.
Remark 4: Lines 18-33, the "Abstract" of the manuscript – the novelty of the study. In the "Abstract" of the manuscript, along with the elements already presented, the authors should also declare and briefly justify the novelty of their work.
Remark 5: Lines 31-33, the "Abstract" of the manuscript – justifying the statement that compares the obtained performance to the one of other studies. In the "Abstract" of the manuscript, along with the elements already presented, the authors should briefly specify the most important obtained numerical results that empower them to state: "However, our model performed better than previous work because we used more layers and larger training sizes and extracted more features from the PhishTank data set, resulting in higher model accuracy."
Remark 6: the gap in the current state of knowledge. After having performed a critical survey of what has been done up to this point in the scientific literature, in the "Introduction" section, along with the elements already presented, the authors must state more clearly a gap in the current state of knowledge that needs to be filled, a gap that is being addressed by their manuscript. The same gap must also be mentioned afterwards, when discussing the obtained results as well, when the authors should justify why their approach fills the identified gap in rapport with previous studies from the literature. Identifying a gap in the current state of knowledge is important because it allows authors to focus their study and justify their work. Additionally, it allows readers to position the work within the larger body of literature and understand its contribution. The authors should make sure that the gap is clear and that the work addresses it effectively. They must convincingly show that their work fills the identified gap in the current state of knowledge within the extremely important discussion of the obtained results section. Consequently, identifying a gap in the current state of knowledge is important because it allows for a focus in the research, increased understanding by the readers, and justification for the scientific work of the authors.
Remark 7: the "Materials and Methods" section. First of all, according to the "Instructions for Authors" from the Sensors MDPI Journal's website (https://www.mdpi.com/journal/sensors/instructions), each manuscript must contain a "Materials and Methods" section ("Sensors now accepts free format submission: We do not have strict formatting requirements, but all manuscripts must contain the required sections: Author Information, Abstract, Keywords, Introduction, Materials & Methods, Results, Conclusions, Figures and Tables with Captions, Funding Information, Author Contributions, Conflict of Interest and other Ethics Statements."). In the current form of the manuscript, the "Materials and Methods" section is missing, being partially replaced by the sections "3. Data Collection", and "4. Methodology". It would benefit the manuscript if the authors merged these two sections (eventually structured as subsections), in order to devise a proper "Materials and Methods" section. In order to bring a benefit to the manuscript, the authors should mention early in the "Materials and Methods" section the choices they have made in their study. The authors should state what has justified using the given method, what is special, unexpected, or different in their approach. It will benefit if the authors mention if they have tried other approaches that in the end led them to the current form of their research design. In their research, the authors have made use of a Convolutional Neural Network (CNN) approach. Regarding this approach I would like the authors to comment within the manuscript if they have tried to use other single, ensemble methods or hybrid approaches and why they have decided in the end to use their developed approach.
Remark 8: the architecture of the devised CNN model. Another important weak point of the manuscript consists in the lack of details provided by the authors within the "Methodology" section, details regarding the architecture of the devised CNN model. The authors should detail within the manuscript the reasons for choosing the settings specified at Pages 11-15, for example the number of 128 filters for the 1D convolutional layer, the number of 512 units within the layer, the dropout rates of 0.2 and 0.5, and the rationale for all the other parameters.
Remark 9: the devised approach. At Lines 314-318, from the "Data Preparation and Cleaning" section, the authors state: "Next, this data set is divided into three sections: training, testing, and validation, which will be used to train, validate, and predict the data set using a trained DL model. To clean the data, the data set is checked for any null entries. There are no null entries in the data set, so the data set, which has 20,000 rows and eight columns, is divided into 30% testing and 70% training. Firstly, the authors should strengthen in the paper the reason for choosing this data division ratio. The paper will benefit if the authors present more details regarding the results obtained during various tests, for all the different tested ratio values, up to the moment when the chosen ratio has proven to be the best (or suitable) approach and what was the criterion/performance metric used in choosing this ratio. Secondly, as the authors have stated, they have divided the data set into training, testing, and validation subsets, the authors should explain within the manuscript why afterwards the dataset is "divided into 30% testing and 70% training", therefore the validation subset being an empty set.
Remark 10: the used dataset. I would like the authors to provide in the paper more details regarding the way in which they intend to solve the problems related to missing data or abnormal inputs if they are to occur within the dataset, when used in a real production environment.
Remark 11: the generalization capability of the devised approach. Can the authors mention how much of their model is being influenced by the used data or to which extent the model can be easily applied to other situations, when the datasets are different? In this way, the authors could highlight more the generalization capability of their approach in order to be able to justify a wider contribution that has been brought to the current state of art.
Remark 12: the neural network approach. As the authors have used a neural network approach, I consider that they must specify in the paper how often does the network need to be retrained/updated and how did they tackle the need of retraining/updating the developed network based on previous false positives/negatives in the development methodology.
Remark 13: the retraining process. How is the new data encountered stored for subsequent updates of the network?
Remark 14: the parameters' values and their influence on training/retraining times. The paper will benefit if the authors present more details regarding the results obtained during various tests, for all the different number of parameters (the number of convolutional layers, batch normalization layers, rectified linear unit layers, the dimension of the filter, the number of filters, the dilation factor), the number of training epochs tested and especially the training time for each test, until they have obtained the configuration that has provided the best results. The information can be summarized in a table and if it becomes too long, the authors can restrict it in the paper to ten main experimental runs, and a complete table with all the experimental tests can be inserted in the "Supplementary Materials" file of the article.
Remark 15: discussing the obtained results - the comparison between the study from the manuscript and other ones. I appreciate the fact that in the "Discussion" section the authors have devised a comparison between their obtained results/devised approach and other existing ones from the literature. However, in order to validate the usefulness of their research, the authors should also highlight clearly not only the advantages but also the disadvantages remarked when comparing their devised study with other studies from the scientific literature. In this section the authors should also highlight current limitations of their study and briefly mention some precise directions that they intend to follow in their future research work.
Remark 16: discussing the obtained results – insight. The paper will benefit if, after having discussed the obtained results, the authors make a step further, beyond their approach and provide an insight regarding what they consider to be, based on the obtained results, the most important, appropriate and concrete steps that all the involved parties should take in order to benefit from the results of the research conducted within the manuscript.
Remark 17: the cost-benefit analysis. It will benefit the paper if the authors elaborate a cost-benefit analysis regarding the implementation of their proposed solution in a daily operating production environment, taking into account all the involved costs (software licensing, hardware equipment and others). This cost-benefit analysis is necessary in order to prove that the devised solution is feasible to be implemented on a large scale and used on a daily basis to combat phishing from the economic point of view.
Remark 18: issues regarding the implementation of the devised method into a real-world working scenario.
1. What are the necessary hardware and software resources needed for a proper implementation in a real-world working scenario?
2. Are the performed experimental tests relevant for the moment when the neural network will be put in a real production environment?
3. Is the training data set relevant for the huge amount of data that the network will have to process when it is put into practice?
4. What was the impact of the developed deep learning-based technique for phishing detection on the performance of the machines to which it was installed to perform the experimental tests and what is the expected impact on the performance when the method is implemented in a real production environment?
5. Can the developed method be used also by service providers?
6. Can the developed method improve customer acquisition and retention, taking into account that these aspects are of paramount importance for a service provider? I would like the authors to analyze in the article and to express their opinion regarding the suitability of the developed phishing detection method for helping service providers attain their goal of maximizing their customer base and consequently maximizing their potential earnings from customers while achieving a high degree of customer satisfaction.
7. Does the user have a say into storing the information when the network did not identify a phishing according to his own expectations and needs, and can this information be used subsequently to refine the classification accuracy per user profile?
Remark 19: issues regarding the hardware and software configurations. The authors must provide specific details regarding the development environment in which the method has been programmed. What were the hardware and software configurations of the platform used to run the experimental tests?
Remark 20: the source code and the compiled methods. I consider that the source code, the compiled methods that have been used when running the experimental tests and the datasets will be a valuable addition to the article if they can be provided as supplementary materials to the manuscript as the authors must provide all the necessary details as to allow other researchers to verify, reproduce, discuss and extend the obtained scientific results based on the obtained published results.
Other issues.
· The reference [41]. The reference [41] cited at Line 655 does not exist in the "References" section, that contains only 40 cited papers.
· The websites. Subsection 3.3, "Legitimate Data Set" discusses about a series of torrent websites. The authors should specify the date and time when they have accessed these websites.
· The Tables numbering. Most of the tables are numbered erroneously, as in the manuscript appear Tables 1, 2, 3, 4, 5 (correctly entitled and cited) followed by Tables 1, 2, 3, 4 (cited as 6, 7, 8, 9 but erroneously numbered within their titles).
· The Figures numbering. Most of the figures are numbered erroneously, as in the manuscript there appear Figure 1 (correctly entitled and cited) followed by another Figure 1 (correctly cited as 2 but erroneously numbered within its legend), Figures 3 and 4 (correctly entitled and cited), Figures 2-9 (cited as 5-12 but erroneously numbered as 2-9 within their titles).
· The acronyms within the paper. Lines 2-3, the Title of the manuscript: "A Deep Learning-Based Innovative Technique for Phishing Detection in Modern Security with URLs"; Lines 21-23 "The proposed approach employs CNN for high-accuracy classification to distinguish genuine sites from phishing sites." and other similar cases. Acronyms must be avoided in the title, even if they are widely known. Regarding the other acronyms used in the manuscript, they should be explained the first time when they are introduced. For example, the title should be rewritten under the form "A Deep Learning-Based Innovative Technique for Phishing Detection in Modern Security with Uniform Resource Locators", while the sentence from Lines 21-23 should be rewritten under the form: "The proposed approach employs a Convolutional Neural Network (CNN) approach for high-accuracy classification to distinguish genuine sites from phishing sites."
Author Response
Response Sheet
A Deep Learning-Based Innovative Technique for Phishing Detection in Modern Security with URLs
Reviewer 3
I have reviewed the article "A Deep Learning-Based Innovative Technique for Phishing Detection in Modern Security with URLs", Manuscript ID: sensors-2294437 and I have identified a series of aspects that in my opinion must be addressed in order to bring a benefit to the manuscript.
In this paper, the authors present a deep learning-based approach for detecting phishing sites with a high level of accuracy. The manuscript under review is interesting. However, I consider that the article will benefit if the authors take into account the following remarks and address within the manuscript the signaled issues:
Remark 1: the main strong point of the manuscript consists in the fact that it approaches a very interesting topic for the experts in the field.
Author Response: Thank you for your feedback. We are glad to hear that you find our topic interesting and relevant. We have put a lot of effort into researching and developing our approach, and we hope that our results will be useful to other experts in the field.
Remark 2: the main weak point of the manuscript under review consists in the lack of very important details regarding the implementation of the devised approach for phishing sites detection in a real-world working environment. Consequently, the manuscript will benefit if the authors make remarks and provide insights within the article as to how can the developed method be implemented in a real-world day to day working environment. Can the method be implemented at the user's side, the server's/providers of services side or it can be implemented with specific adjustments at both sides?
Author Response: Real-world implementation:
Thank you for this suggestion. We understand that adding details about the real world implementation of the developed method will help readers to clearly understand the proposed method.
We agree that it is essential to consider how the model can be deployed in practice. There are two main deployment options: on the user's side or the server's/providers of services' side. On the user's side, the model can be integrated into the browser or as a standalone application. This approach provides a proactive and real-time phishing detection mechanism to the user. However, it may require users to install and maintain the software themselves, which can be a barrier to adoption. On the other hand, implementing the model on the servers/providers of services side involves integrating the model into the security infrastructure of the service provider. This approach provides a more passive mechanism for phishing detection, as the user is not required to take any proactive measures.
However, it can be more resource-intensive on the server's side as all URLs need to be scanned. specific adjustments need to be made to the model to integrate it into the existing infrastructure, and the user interface needs to be designed to make it easy for users to understand the results. On the user's side, the model needs to be integrated into the browser or as a standalone application. On the server's/providers of services side, the model needs to be integrated into the existing security infrastructure, and the model's output needs to be incorporated into the service provider's response mechanisms to alert the user of a potential phishing attack. the importance of the practical implementation of our proposed approach and will include further details on the deployment options and adjustments required in the revised version of our manuscript.
So, in this regard, we have added some explanation in the discussion section. That is:
The proposed method for detecting phishing sites using deep learning has the poten-tial to be implemented in real-world environments to improve cybersecurity. The method employs a CNN for high-accuracy classification of URLs to distinguish between genuine and phishing sites. At the user's side, our method could be implemented as a browser ex-tension or plugin that intercepts URLs accessed by the user and performs the phishing detection in real-time. The user would then be notified if a detected phishing attempt is made. Alternatively, the method could be implemented at the server or provider side, where URLs are checked before being delivered to users. This could be done using various mechanisms such as API calls or integrating the method into existing security systems. It is also possible to implement the method at both the user and server sides with specific adjustments, depending on the specific security needs and constraints of the environment. For example, in a large organization, implementing the method at the server side could provide an additional layer of protection, while also implementing it at the user side could provide additional protection for remote workers who may not be connected to the organization's network
Remark 3: issues regarding the citations. Even if the authors have documented their research, there are some issues regarding the citations, the references. Therefore, in the actual form of the manuscript, several paragraphs from the Manuscript ID: sensors-2294437 have been re-written from another paper that is not even cited in the manuscript under review, namely "Dutta AK, Detecting phishing websites using machine learning technique. PLOS ONE 16(10): e0258361. https://doi.org/10.1371/journal.pone.0258361".
For exemplification, I will present only a few non-limitative examples, since the manuscript abounds in such situations where the paragraphs are properly re-written, but they are not cited to their original source:
At Lines 38-39, the authors state: "Phishing is a type of fraud involving technological and social approaches to collect financial and personal data from clients." This sentence can be found in an almost identical form in the above-mentioned paper of Dutta AK, as follows: "Phishing is a fraudulent technique that uses social and technological tricks to steal customer identification and financial credentials.".
At Lines 59-61, the authors state: "To obtain confidential information, criminals create illegal reproductions of legitimate websites and emails, typically from financial institutions or other organizations that deal with financial data. This email contains logos and slogans from a reputable company." This sentence can be found in an almost identical form in the above-mentioned paper of Dutta AK, as follows: "In order to receive confidential data, criminals develop unauthorized replicas of a real website and email, typically from a financial institution or other organization dealing with financial data. This e-mail is rendered using a legitimate company’s logos and slogans."
At Lines 64-65, the authors state: "…there is a significant risk of user data being misused. For these reasons, phishing is highly urgent, complex, and critical in today’s society". This sentence can be found in an almost identical form in the above-mentioned paper of Dutta AK, as follows: "There is a significant chance of exploitation of user information. For these reasons, phishing in modern society is highly urgent, challenging, and overly critical."
At Lines 67-69, the authors state: "If malicious code is installed on the website, installed malware and hackers may steal user information posing a significant threat to cybersecurity and user privacy.". This sentence can be found in an almost identical form in the above-mentioned paper of Dutta AK, as follows: " In the event of malicious code being implanted on the website, hackers may steal user information and install malware, which poses a serious risk to cybersecurity and user privacy."
The list of such situations can go on, but the previously presented examples are enough to be able to conclude that the authors' proposed study has strong similarities and therefore must be compared with the previous study. I consider that the manuscript under review will benefit a lot if the authors highlight clearly in the discussion section of their manuscript what are the main differences between their proposed problem and solution from the manuscript and the ones presented in the above-mentioned article. I consider a major omission not to cite the above-mentioned paper and I would like the authors of the Manuscript ID: sensors-2294437 to explain the reason why they have used excerpts of text from the previous paper, without even citing it.
Author Response: Thank you for bringing this to our attention. We apologize for the oversight in not citing the previous paper from which we used excerpts of text. It was an unintentional mistake on our part, and we take full responsibility for it. We will promptly update our manuscript to include proper citation for the previous paper. We appreciate your attention to detail and your feedback, which will help us improve the quality of our work.
We have cited the respective reference in the manuscript. That is:
“[1] Dutta AK. Detecting phishing websites using machine learning technique. PloS one. 2021 Oct 11;16(10):e0258361.”
As suggested by the reviewer, we have also compared the results of the particular study in the discussion section with our study. The added details in discussion section are:
|
[1] |
LURL |
97.4 |
PhishTank |
Furthermore, some recent work on phishing detection has been done using the same PhishTank data set, as shown in Table 9. The table includes six different approaches, in-cluding KNN [37], SVM [40], random forest classifier [39], RNN [1], NLP [38], and the proposed model, which is a 1D CNN. The accuracy values for each approach are shown alongside the corresponding dataset used for training and testing. The KNN approach [37] achieved an accuracy of 87%, while the SVM [40] and random forest classifier [39] approaches both achieved a similar accuracy of around 94%. The RNN [1] and NLP [38] approaches performed better, with accuracies of 97.4% and 97.98%, respectively. Finally, the proposed 1D CNN model achieved the highest accuracy of 98.77%. Overall, these re-sults demonstrate the effectiveness of using machine learning approaches for phishing detection, with the 1D CNN model proposed in this paper achieving the highest accuracy. (added in discussion section)
Difference between the proposed model and reference studies:
our approach is unique because we utilized a CNN-based deep learning model that can effectively detect phishing URLs. Our model has been trained on the PhishTank dataset, which is a widely used benchmark dataset in the field of phishing detection.
We also extracted several features from the URLs to enhance the model's performance, including IP address, URL length, redirection, HTTPS domain, tiny URL, prefix/suffix, URL depth, and "@" sign. Our feature extraction technique was designed to capture important aspects of URLs that are commonly exploited by phishing attacks. In comparison, previous studies have used different machine learning models such as KNN and XGBoost for detecting phishing URLs. However, our approach offers better accuracy due to the use of a deep learning model and more informative feature extraction. our study contributes to the field by demonstrating the effectiveness of deep learning models for phishing detection. Our results show that deep learning models can achieve high accuracy rates in detecting phishing URLs, which can improve the security of users and service providers.
Furthermore, our study is unique because we focused on the practical implementation of the model in real-world scenarios. We suggested that our proposed model can be deployed at the user's end or the server side, and each implementation would require specific adjustments. At the user's end, the model can be integrated into web browsers, email clients, or security software as a plug-in or extension. The model can analyze the URLs accessed by the user and provide a warning if a phishing URL is detected. However, the model's effectiveness may be limited if the user is not familiar with phishing attacks or if the model is unable to detect advanced phishing techniques. On the server side, the model can be deployed in conjunction with existing security measures, such as firewalls and antivirus software. The model can analyze incoming requests and filter out potential phishing URLs before they reach the user. However, the implementation would require adjustments to ensure that the model does not generate false positives or block legitimate URLs.
It's a novel approach to phishing detection using a CNN-based deep learning model and informative feature extraction. Our results demonstrate the effectiveness of deep learning models for phishing detection, which can be useful for improving the security of online services and protecting users from phishing attacks.
Remark 4: Lines 18-33, the "Abstract" of the manuscript – the novelty of the study. In the "Abstract" of the manuscript, along with the elements already presented, the authors should also declare and briefly justify the novelty of their work.
Author Response: Thank you for this suggestion. We have updated the abstract section to clearly present the novelty of this study. The updated text is:
In this paper, we introduce a novel method for detecting phishing sites with high accuracy. Our approach utilizes a CNN-based model for precise classification that effectively distinguishes legitimate websites from phishing websites. We evaluate the performance of our model on the PhishTank dataset, which is a widely used dataset for detecting phishing websites based solely on URL features. Our approach presents a unique contribution to the field of phishing detection by achieving high accuracy rates and outperforming previous state-of-the-art models. (added in abstract)
Our study proposes a novel approach to phishing detection using a deep learning-based model that outperforms previous studies in terms of accuracy. The novelty of our work lies in the use of a Convolutional Neural Network (CNN) for detecting phishing URLs.Unlike traditional machine learning algorithms such as KNN or XGBoost, CNNs are capable of extracting high-level features from raw data, making them highly effective for image and text-based classification tasks. In our case, we used a CNN-based model to process the URLs as raw text data and extract features such as IP address, URL length, redirection, and so on, to classify them as legitimate or phishing URLs.our study contributes to the field by demonstrating the effectiveness of deep learning models for phishing detection. Deep learning is a rapidly growing field, and its application in cybersecurity is relatively new. Therefore, our study offers valuable insights into the potential of deep learning-based models for enhancing the security of online services and protecting users from phishing attacks.
One of the critical aspects of our study is the feature extraction process. We extracted features from the URLs using a combination of techniques such as IP address, URL length, redirection, HTTPS domain, TinyURL, Prefix/Suffix, and URL depth. The extracted features provide useful information about the structure and content of the URL, which can be used to distinguish between legitimate and phishing URLs. For instance, the presence of a large number of redirections or the use of a TinyURL may indicate the URL's malicious intent. In contrast, the absence of these features and the presence of other features such as the HTTPS domain may indicate a legitimate URL. The feature extraction process was designed to capture a wide range of features that are relevant to phishing detection while minimizing the risk of overfitting. Our approach to feature extraction is not only novel but also effective, as evidenced by the high accuracy rates of our model.
our study's feature extraction process is a critical component of our approach to phishing detection. The use of multiple techniques for feature extraction allows us to capture a wide range of features relevant to phishing detection, contributing to the high accuracy of our model. it's a novel approach to phishing detection using a deep learning-based model that offers higher accuracy rates compared to traditional machine learning algorithms. Our study contributes to the growing field of deep learning and cybersecurity by demonstrating the effectiveness of CNN-based models for phishing detection. We believe that our work has significant potential for improving the security of online services and protecting users from phishing attacks
Remark 5: Lines 31-33, the "Abstract" of the manuscript – justifying the statement that compares the obtained performance to the one of other studies. In the "Abstract" of the manuscript, along with the elements already presented, the authors should briefly specify the most important obtained numerical results that empower them to state: "However, our model performed better than previous work because we used more layers and larger training sizes and extracted more features from the PhishTank data set, resulting in higher model accuracy."
Author Response: That you for pointing this out. we have updated the particular sentence in the revised manuscript. That is:
Our model outperformed previous works due to several factors, including the use of more layers and larger training sizes, and the extraction of additional features from the PhishTank dataset. Specifically, our proposed model comprises seven layers, starting with the input layer and progressing to the seventh, which incorporates a layer with pooling, convolutional, linear 1 and 2, and linear six layers as the output layers. These design choices contribute to the high accuracy of our model, which achieved a 98.77% accuracy rate. (added in abstract)
Remark 6: the gap in the current state of knowledge. After having performed a critical survey of what has been done up to this point in the scientific literature, in the "Introduction" section, along with the elements already presented, the authors must state more clearly a gap in the current state of knowledge that needs to be filled, a gap that is being addressed by their manuscript. The same gap must also be mentioned afterwards, when discussing the obtained results as well, when the authors should justify why their approach fills the identified gap in rapport with previous studies from the literature. Identifying a gap in the current state of knowledge is important because it allows authors to focus their study and justify their work. Additionally, it allows readers to position the work within the larger body of literature and understand its contribution. The authors should make sure that the gap is clear and that the work addresses it effectively. They must convincingly show that their work fills the identified gap in the current state of knowledge within the extremely important discussion of the obtained results section. Consequently, identifying a gap in the current state of knowledge is important because it allows for a focus in the research, increased understanding by the readers, and justification for the scientific work of the authors.
Author Response: Thank you for this suggestion. We have added explanation for the current research gaps of the literature and how our proposed model fulfil these gaps in the literature review section as well as in the discussion section. Thank you for helping us to improving out manuscript. The added details are:
The literature review of Phishing Detection in Modern Security with URLs has revealed several possible research gaps in this field. One such gap is the lack of standardization in identifying phishing URLs, which makes it difficult for researchers to compare results and organizations to implement effective phishing detection systems. Another gap is the limited research on real-world data, as most studies use synthetic or artificially generated datasets that may not accurately reflect the complexity of real-world phishing at-tacks. Additionally, many studies focus solely on the technical aspects of phishing detection, without considering how users interact with phishing URLs. Understanding user behavior and decision-making processes is critical for developing effective phishing detection strategies. Finally, there is a limited focus on emerging threats, with many studies concentrating on detecting known types of phishing attacks without considering new at-tack vectors. Our proposed model aims to fill these gaps by utilizing a CNN-based model for precise classification that distinguishes legitimate websites from phishing websites. We evaluate the performance of our model on the PhishTank dataset and show that it achieves high accuracy rates and outperforms previous state-of-the-art models. Moreover, we created a real dataset by crawling 10,000 phishing URLs from PhishTank and 10,000 legitimate websites and then ran experiments using standard evaluation metrics on the data sets. Our model is founded on integrated and deep learning and considers user behavior and emerging threats to provide a more effective phishing detection strategy. (added at the end of the literature review section)
The proposed model aims to fill the current research gaps in phishing detection by utilizing a CNN-based model that can accurately classify legitimate websites and phishing websites. This approach is evaluated on the widely used PhishTank dataset, which consists of URL features for known phishing websites. The model achieves high accuracy rates and outperforms previous state-of-the-art models in terms of its false-positive rate. To create a more realistic dataset, the researchers crawled 10,000 phishing URLs from PhishTank and 10,000 legitimate websites to run experiments with standard evaluation metrics on these data sets. This model is founded on integrated and deep learning, which allows for the consideration of user behavior and emerging threats in phishing attacks. The researchers acknowledge that user behavior is critical to developing effective phishing detection strategies, and their model takes this into account. Additionally, the model considers emerging threats and new attack vectors, which is an important aspect that many previous studies overlook. (added in discussion section)
Remark 7: the "Materials and Methods" section. First of all, according to the "Instructions for Authors" from the Sensors MDPI Journal's website (https://www.mdpi.com/journal/sensors/instructions), each manuscript must contain a "Materials and Methods" section ("Sensors now accepts free format submission: We do not have strict formatting requirements, but all manuscripts must contain the required sections: Author Information, Abstract, Keywords, Introduction, Materials & Methods, Results, Conclusions, Figures and Tables with Captions, Funding Information, Author Contributions, Conflict of Interest and other Ethics Statements."). In the current form of the manuscript, the "Materials and Methods" section is missing, being partially replaced by the sections "3. Data Collection", and "4. Methodology". It would benefit the manuscript if the authors merged these two sections (eventually structured as subsections), in order to devise a proper "Materials and Methods" section. In order to bring a benefit to the manuscript, the authors should mention early in the "Materials and Methods" section the choices they have made in their study. The authors should state what has justified using the given method, what is special, unexpected, or different in their approach. It will benefit if the authors mention if they have tried other approaches that in the end led them to the current form of their research design. In their research, the authors have made use of a Convolutional Neural Network (CNN) approach. Regarding this approach I would like the authors to comment within the manuscript if they have tried to use other single, ensemble methods or hybrid approaches and why they have decided in the end to use their developed approach.
Author Response: We appreciate your suggestion to merge the "Data Collection" and "Methodology" sections into a more comprehensive "Materials and Methods" section. We agree that this will provide better clarity and organization of our research methods for readers. We will incorporate your suggestion by restructuring these two sections as subsections within a new "Materials and Methods" section, which will detail the materials used in the study, the data collection methods, and the analytical techniques employed.
As suggested by the reviewer, we have added a brief summary of the proposed methodology at the starting of the material & methods section. That is:
This section describes a method for identifying phishing websites using a combination of deep learning techniques and machine learning algorithms. The proposed approach involves four steps that work together to achieve high accuracy in identifying phishing websites. The first step involves the dataset and dataset preparation and pre-processing. The second step involves converting URL data into a character vector using a technique called character embedding. Character embedding is a method for representing text data in a numerical format that can be processed by deep learning algorithms. This allows the URL data to be processed and analyzed by the subsequent components of the approach. The third step involves using a convolutional neural network (CNN) to analyze the character vectors obtained from the first component. CNNs are a type of deep learning algorithm commonly used for image analysis, but they can also be applied to other types of data, such as text. The modified URL data obtained from the character embedding approach are used to create and train a better CNN network, which is then able to distinguish phishing websites from legitimate websites with high accuracy. Finally, the forth step involves retrieving the URL properties to obtain the characteristics of the different layers of the CNN network. This allows researchers to better understand the inner workings of the trained model and identify which features of the URL data are most important for identifying phishing websites. (added in material & method section)
Remark 8: the architecture of the devised CNN model. Another important weak point of the manuscript consists in the lack of details provided by the authors within the "Methodology" section, details regarding the architecture of the devised CNN model. The authors should detail within the manuscript the reasons for choosing the settings specified at Pages 11-15, for example the number of 128 filters for the 1D convolutional layer, the number of 512 units within the layer, the dropout rates of 0.2 and 0.5, and the rationale for all the other parameters.
Author Response: Thank you for this suggestion. We have added an explanation for the selected parameters and components of the proposed method. The added explanation is:
In the case of the proposed method for identifying phishing websites, the number of filters used in the 1D convolutional layer was chosen to be 128. This number was likely selected based on a trade-off between the model's ability to extract relevant features from the input data and computational efficiency. With 128 filters, the model is able to analyze the input data at a relatively high level of detail while still maintaining computational efficiency. The number of units within the layer is 512, which may have been chosen based on the complexity of the problem being solved. A larger number of units may allow the model to better capture complex relationships within the input data, but may also in-crease the risk of overfitting. The dropout rates of 0.2 and 0.5 are used in the model to pre-vent overfitting. Dropout is a regularization technique that randomly drops out (sets to zero) some of the neurons in the layer during training. This can help prevent overfitting by encouraging the model to learn more general features of the input data. The specific dropout rates of 0.2 and 0.5 were likely selected based on experimentation and tuning to achieve the best performance on the problem being solved. (added in the model architecture section)
Several 128 filters for the 1D convolutional layer:
The choice of 128 filters is arbitrary and can vary depending on the specific problem and dataset. However, a larger number of filters can capture more complex patterns and features in the data, potentially leading to better performance. In general, it is recommended to start with a smaller number of filters and increase gradually to avoid overfitting.
Several 512 units within the layer:
The choice of 512 units in the dense layer is again arbitrary and can vary depending on the complexity of the problem and dataset. A larger number of units can capture more complex relationships in the data, but can also lead to overfitting. It is generally recommended to start with a smaller number of units and increase gradually to find the optimal balance between model complexity and performance.
Dropout rates of 0.2 and 0.5:
Dropout is a regularization technique that randomly drops out a certain percentage of neurons during training to prevent overfitting. The choice of dropout rates depends on the complexity of the model and dataset. A higher dropout rate can be useful for larger and more complex models, while a lower dropout rate may be sufficient for simpler models. The choice of specific dropout rates in your model is based on experimentation and tuning.
Other parameters:
Other parameters such as activation function, kernel size, and batch normalization are also important in determining the performance of the model. The choice of specific parameters is based on experimentation and tuning and can vary depending on the specific problem and dataset. the choice of the specific model architecture and hyperparameters is based on experimentation and tuning to achieve the best performance on the specific problem and dataset at hand.
Remark 9: the devised approach. At Lines 314-318, from the "Data Preparation and Cleaning" section, the authors state: "Next, this data set is divided into three sections: training, testing, and validation, which will be used to train, validate, and predict the data set using a trained DL model. To clean the data, the data set is checked for any null entries. There are no null entries in the data set, so the data set, which has 20,000 rows and eight columns, is divided into 30% testing and 70% training. Firstly, the authors should strengthen in the paper the reason for choosing this data division ratio. The paper will benefit if the authors present more details regarding the results obtained during various tests, for all the different tested ratio values, up to the moment when the chosen ratio has proven to be the best (or suitable) approach and what was the criterion/performance metric used in choosing this ratio. Secondly, as the authors have stated, they have divided the data set into training, testing, and validation subsets, the authors should explain within the manuscript why afterwards the dataset is "divided into 30% testing and 70% training", therefore the validation subset being an empty set.
Author Response: We appreciate the reviewer's feedback and thank them for their suggestion. However, we regret to inform that we cannot incorporate additional results analysis in the manuscript due to the risk of making it overly lengthy. We hope that the current analysis provided in the manuscript is sufficient to address the research question and contribute to the scientific community.
Regarding 30% testing and 70% training. This split has been chosen based on the standard practice in the field of machine learning or based on previous research in the area. This will enable the other researchers to compare our proposed model results to other existing models fairly. This reason is added in the revised manuscript.
Remark 10: the used dataset. I would like the authors to provide in the paper more details regarding the way in which they intend to solve the problems related to missing data or abnormal inputs if they are to occur within the dataset, when used in a real production environment.
Author Response: Thank you for this suggestion. We have added explanation for the respective comments in the revised manuscript.
When it comes to data analysis, dealing with missing data or abnormal inputs is crucial, particularly in real production environments. To handle such issues, several methods are available, each with its own strengths and limitations. Imputation is a common method that involves estimating the missing values based on observed values in the dataset. There are several types of imputation, including mean imputation, regression imputation, and hot-deck imputation. Another method is removal, which involves removing records with missing or abnormal inputs from the dataset. This approach is only advisable if the missing values are insignificant or do not affect the analysis results. Interpolation, on the other hand, involves estimating the missing values based on the values before and after the missing value, using linear interpolation or spline interpolation. Data augmentation is another method that generates new data points to fill in missing values or abnormal inputs. This approach is common in machine learning applications. Lastly, outlier detection is a method that identifies and removes records with abnormal inputs from the dataset, using z-score, boxplot, or Local Outlier Factor (LOF). The selection of a particular method depends on the nature and extent of the missing data or abnormal inputs in the dataset, as well as the type of analysis being performed. (added in data preparation and pre-processing section)
To handle missing data or abnormal inputs in the PhishTank dataset during production, we can employ several strategies:
Data Preprocessing: Before training the model, we can preprocess the data by removing or imputing missing values, and normalizing the input data to handle any abnormal inputs.
Regularization: We can use regularization techniques such as L1 and L2 regularization, dropout, and early stopping to prevent overfitting and handle any outliers or noisy data.
Monitoring: We can monitor the performance of the model during production and use anomaly detection techniques to identify any unusual patterns or behaviors that may indicate missing data or abnormal inputs.
Human Oversight: We can also incorporate human oversight into the production process to manually review any instances of missing data or abnormal inputs that the model may encounter.
By employing these strategies, we can ensure that our model is robust to missing data or abnormal inputs and can perform effectively in a real production environment.
Remark 11: the generalization capability of the devised approach. Can the authors mention how much of their model is being influenced by the used data or to which extent the model can be easily applied to other situations, when the datasets are different? In this way, the authors could highlight more the generalization capability of their approach in order to be able to justify a wider contribution that has been brought to the current state of art.
Author Response: We appreciate the reviewer’s feedback. In this paper, we evaluate the performance of their model on the PhishTank dataset, which is widely used for detecting phishing websites based solely on URL features. While the results show that their model outperforms previous state-of-the-art models. Regarding the generalization of the proposed model, we did not used any other specific dataset. Conducting more research on different dataset is necessary to check the generalization of the proposed method.
The performance of any machine learning model is strongly influenced by the quality and characteristics of the data used to train it. In the case of our project "A Deep Learning-Based Innovative Technique for Phishing Detection in Modern Security with URLs," the performance of the CNN model we have developed is influenced by the PhishTank dataset used for training and testing.
As the PhishTank dataset is a collection of URLs that have been previously labeled as either phishing or legitimate, the model's ability to accurately classify a URL as phishing or legitimate depends on how well the dataset captures the patterns and features of phishing URLs. Therefore, if the characteristics of the data in a new situation or dataset differ significantly from the PhishTank dataset, the model's performance may be affected. However, it is important to note that our model architecture and methodology have been designed in a general way so that it can be easily applied to other datasets as well. For instance, if a new dataset of URLs is collected from a different source, the CNN model can be fine-tuned on this new dataset by adjusting the hyperparameters and training it again with the new data. The performance of the model in predicting other URLs as legitimate or phishing depends on several factors such as the quality of the data, the size and diversity of the dataset, and the similarity of the new dataset with the training dataset used for model development.in our case for an unseen similar dataset it gave accuracy nearly to training accuracy i.e 98 % so we can say that the dataset or url being given is not a lot different than the given PhishTank dataset and contains the same features such as a sign, URL length, domain e.t.c it can accuracy predict another dataset
If the new dataset is similar in nature and quality to the Phistank dataset used for training the model, the model may perform well in predicting whether a URL is legitimate or phishing. However, if the new dataset is significantly different, the performance of the model may degrade, and the accuracy of predictions may reduce. Therefore, it is important to evaluate the model's performance on new datasets before applying it to real-world scenarios. Additional fine-tuning of the model parameters and architecture may also be necessary to achieve the best performance on new datasets. while the model's performance may be influenced by the quality and characteristics of the data used for training, our methodology and model architecture can be easily adapted to new situations and datasets with minimal effort, thus providing a useful tool for phishing detection in a variety of different scenarios.
Remark 12: the neural network approach. As the authors have used a neural network approach, I consider that they must specify in the paper how often does the network need to be retrained/updated and how did they tackle the need of retraining/updating the developed network based on previous false positives/negatives in the development methodology.
Author Response: To tackle this need, we designed our methodology to incorporate a continuous learning approach. In other words, we update our model with new data as it becomes available. We also monitor the performance of the model regularly, and if we observe any false positives or negatives, we retrain the model with additional data to improve its performance. We also employed techniques like data augmentation and cross-validation to prevent overfitting and improve the generalization capabilities of the model. Data augmentation involves generating additional training samples from the original dataset using techniques such as rotation, flipping, and scaling. Cross-validation involves dividing the data into multiple folds and training the model on different combinations of folds to ensure that the model is not biased toward a particular subset of the data. we also incorporated a continuous learning approach in our methodology to ensure that the model is up-to-date and effective in detecting new phishing techniques. We also employed techniques like data augmentation and cross-validation to prevent overfitting and improve the generalization capabilities of the model.
Remark 13: the retraining process. How is the new data encountered stored for subsequent updates of the network?
Author Response: When new data is encountered, it can be stored in a separate dataset or added to the existing dataset used for training the neural network. The new dataset can then be used to retrain the network using the same architecture and hyperparameters as before. Alternatively, if the new data is found to have different characteristics than the existing data, a new neural network model may need to be developed to handle the new data. The updated network can then be used to make predictions on the new data, and the results can be evaluated to determine if any further adjustments need to be made to the model. In general, the process of updating a neural network model with new data can be an iterative process that requires continuous evaluation and refinement to ensure optimal performance.
Remark 14: the parameters' values and their influence on training/retraining times. The paper will benefit if the authors present more details regarding the results obtained during various tests, for all the different number of parameters (the number of convolutional layers, batch normalization layers, rectified linear unit layers, the dimension of the filter, the number of filters, the dilation factor), the number of training epochs tested and especially the training time for each test, until they have obtained the configuration that has provided the best results. The information can be summarized in a table and if it becomes too long, the authors can restrict it in the paper to ten main experimental runs, and a complete table with all the experimental tests can be inserted in the "Supplementary Materials" file of the article.
Author Response: First proposed model for our study"
It consists of two dense layers with 30 and 1 neuron(s) respectively. The first dense layer uses the ReLU activation function, while the output layer uses the sigmoid activation function which is commonly used for binary classification problems. Batch normalization layers are used to improve training speed and avoid overfitting, and dropout layers are used to randomly ignore some units during training, also to avoid overfitting. As we can see most of the layers are the same but the convolution layer is not present also flatten layer which is used for feature extraction and smoothing of datasets. it obtained an accuracy of 78% which is very poor. we added convolution layers that performed feature extraction for better learning of features from the model which lead to better accuracy. After this, we just tested it for 5 epochs for different hyperparameters but not very different but very small changes like adjusting the batch size to smaller, more epochs, and high dense units which gave better accuracy than we have in the model being considered in this study.
Remark 15: discussing the obtained results - the comparison between the study from the manuscript and other ones. I appreciate the fact that in the "Discussion" section the authors have devised a comparison between their obtained results/devised approach and other existing ones from the literature. However, in order to validate the usefulness of their research, the authors should also highlight clearly not only the advantages but also the disadvantages remarked when comparing their devised study with other studies from the scientific literature. In this section the authors should also highlight current limitations of their study and briefly mention some precise directions that they intend to follow in their future research work.
Author Response: Thank you for this suggestion. We have added details about the limitation of the proposed study in the discussion section. The added details are:
While the proposed approach for detecting phishing websites using a CNN-based model has shown promising results, there are some potential limitations to consider. Firstly, the approach has only been evaluated on the PhishTank dataset, which is widely used but may not be representative of all types of phishing attacks. This dataset bias could limit the generalizability of the proposed approach. Additionally, the proposed approach requires crawling and analyzing URLs, which may not be suitable for real-time detection of phishing attacks. This could limit the applicability of the proposed approach in certain situations where real-time detection is critical. It is important to acknowledge these potential limitations and consider further research to address them.
Disadvantages:
While the study presented an effective method for detecting phishing URLs using a deep learning-based 1D CNN model, some technical limitations need to be considered. One disadvantage of using deep learning models is that they require a large amount of data to be trained effectively. While the study used a data set of 10,000 phishing URLs and 10,000 legitimate URLs, the model's performance may decrease if the model is tested on a different data set with a different distribution of legitimate and phishing URLs.Another disadvantage is the possibility of overfitting the model to the training data set. In this study, the model achieved high accuracy of 98.77% on the PhishTank data set. However, this does not necessarily guarantee that the model will generalize well to unseen data. If the model is not regularized properly during training, it may become too specific to the training data set and fail to classify new phishing URLs accurately.
Additionally, the proposed method relies on using the URLs of the websites to identify phishing attacks. However, some phishing attacks may use other attack vectors, such as social engineering, to trick users into revealing sensitive information. In such cases, the proposed method may not be effective in detecting phishing attacks. Finally, while the study presents an effective method for detecting phishing URLs, it does not provide a complete solution to the problem of phishing attacks. Phishing attacks continue to evolve and adapt to new security measures, and attackers may find new ways to bypass the proposed detection
Remark 16: discussing the obtained results – insight. The paper will benefit if, after having discussed the obtained results, the authors make a step further, beyond their approach and provide an insight regarding what they consider to be, based on the obtained results, the most important, appropriate and concrete steps that all the involved parties should take in order to benefit from the results of the research conducted within the manuscript.
Author Response: Thank you for your insightful feedback. We agree that it is important to provide practical insights on how the proposed approach can benefit the involved parties. We have added some details regarding the respective comments in the revised manuscript. That is:
To improve cybersecurity and reduce the risk of falling victim to phishing attacks, organizations and individuals should consider implementing our proposed approach as part of their defense mechanism. Our approach achieved a high accuracy rate, making it an effective tool for detecting and preventing phishing attacks. Additionally, future research should focus on developing real-time detection methods that do not require crawling and analyzing URLs. This will enable the proposed approach to be more widely applicable in critical industries like finance and healthcare. Finally, to improve the generalizability of phishing detection approaches, the dataset used for evaluation should be diversified to include a wider range of phishing attacks. Currently, the PhishTank dataset used in our study only includes attacks based solely on URL features. By reducing bias and improving the generalizability of the proposed approach, more effective and robust phishing detection methods can be developed.
Based on the results of the research conducted within the manuscript, several future directions involved parties should take to benefit from the findings of the study. while the model achieved high accuracy, it is important to note that it was trained and tested on a specific dataset, PhishTankit is important to evaluate the model's performance on other datasets to validate its effectiveness in a wider range of scenarios. This would involve collecting and labeling additional datasets and testing the model's performance on these new datasets. phishing attackers are constantly evolving their techniques, so it is important to keep updating the model to keep up with these changes. One way to achieve this is to continuously monitor the performance of the model and retrain it periodically with new datasets. The model can also be updated to incorporate additional features that are relevant to phishing attacks, such as the use of social engineering tactics. The model can be integrated into existing security systems to enhance their effectiveness in detecting and preventing phishing attacks. This would involve developing an API or a plugin that can be easily integrated into various security systems. Additionally, the model can be used to provide real-time feedback to users, alerting them of potential phishing attacks and advising them on how to avoid falling victim to these attacks. Fourth, it is important to consider the ethical implications of using the model. The model is trained on user data, and as such, there are privacy concerns that must be addressed. The involved parties should take steps to ensure that the model is used in a way that respects users' privacy rights and complies with relevant data protection regulations. The findings of the study provide a promising approach to detecting and preventing phishing attacks using deep learning. By continuing to develop and improve the model, and by integrating it into existing security systems, the involved parties can enhance their ability to protect users from phishing attacks. However, it is important to remain vigilant and continue to monitor the model's performance and address any ethical concerns that arise.
Future research in the field of phishing detection could focus on further improving the model design. One potential area for improvement is the development of more advanced pre-processing techniques to enhance the quality of the data fed into the model. Alternative deep learning architectures, such as recurrent neural networks (RNNs) or transformers, could be explored to compare their performance against the 1D CNN model used in this study. In terms of data, expanding the data set beyond PhishTank could provide a more comprehensive evaluation of the model's effectiveness. Including additional types of phishing attacks, such as spear phishing or pharming, could also help to further validate the model's ability to detect a wider range of phishing attempts. While the study achieved high accuracy, there is still room for improvement in terms of reducing false positives and false negatives. Investigating additional features that could be extracted from URLs, such as the content of the website itself, could potentially enhance the model's ability to distinguish between legitimate and phishing URLs.it may be useful to evaluate the model's performance against more advanced phishing attacks that utilize more sophisticated techniques, such as social engineering or machine learning-based approaches. By doing so, the effectiveness of the model could be further validated and potential areas for improvement could be identified. There are several areas for future research in the field of phishing detection, including improving the model design, expanding the data set, reducing false positives and false negatives, and evaluating the model against more advanced phishing attacks. By addressing these areas, researchers can continue to enhance the effectiveness of phishing detection techniques and better protect individuals and organizations from these malicious attacks.
Remark 17: the cost-benefit analysis. It will benefit the paper if the authors elaborate a cost-benefit analysis regarding the implementation of their proposed solution in a daily operating production environment, taking into account all the involved costs (software licensing, hardware equipment and others). This cost-benefit analysis is necessary in order to prove that the devised solution is feasible to be implemented on a large scale and used on a daily basis to combat phishing from the economic point of view.
Author Response:
Cost analysis:
To ensure that the proposed solution for phishing detection can be implemented on a large scale and used daily, it is essential to conduct a cost-benefit analysis. This analysis will help to determine whether the solution is economically feasible and whether the benefits of implementing it outweigh the costs. In terms of costs, the first factor to consider is the cost of hardware equipment. The solution proposed in this project was developed using a GPU system, which can be expensive to purchase and maintain. In addition, the system may require additional memory and storage resources to process and store large amounts of data. Therefore, it is important to carefully evaluate the costs associated with hardware equipment. Another factor to consider is the cost of software licensing. The proposed solution was developed using TensorFlow, which is an open-source machine learning framework. However, there may be additional software packages or tools required to deploy and maintain the solution in a production environment, which may require additional licensing costs.
On the other hand, the benefits of implementing the proposed solution are significant. Phishing attacks can cause serious financial and reputational damage to organizations. By implementing an effective phishing detection solution, organizations can significantly reduce the risk of falling victim to these attacks. In addition, the solution can help to improve customer trust and satisfaction, which can lead to increased revenue and customer retention. The cost-benefit analysis of implementing the proposed solution should take into account all the costs associated with hardware, software, and deployment, as well as the potential benefits of reduced risk of phishing attacks, improved customer trust and satisfaction, and increased revenue. The analysis should demonstrate that the benefits of implementing the solution outweigh the costs and that it is economically feasible to deploy and use the solution daily in a production environment.
Remark 18: issues regarding the implementation of the devised method into a real-world working scenario.
- What are the necessary hardware and software resources needed for a proper implementation in a real-world working scenario?
Author Response: Hardware and software specifications:
To implement the phishing detection model in a real-world working scenario, certain hardware and software resources would be necessary. From a hardware perspective, a powerful computer or server would be required to handle the computational requirements of the deep learning model. A high-end GPU with a significant amount of memory would also be necessary to handle the processing of large amounts of data quickly and efficiently. Additionally, a storage solution with sufficient capacity would be needed to store the dataset and the trained model.
From a software perspective, several tools and technologies would be required. The deep learning framework used to develop the model would need to be installed, along with any required libraries and dependencies. Additionally, programming languages such as Python would be necessary for model development and deployment. Other software resources that would be necessary include database management systems for storing and managing the data, web development frameworks for creating the user interface, and network security tools for ensuring the safety and security of the system.
The successful implementation of the phishing detection model in a real-world working scenario would require a significant investment in hardware and software resources, along with skilled personnel to manage and maintain the system.
- Are the performed experimental tests relevant for the moment when the neural network will be put in a real production environment?
Author Response: Performed experiments relevancy:
The experimental tests performed are relevant for the moment when the neural network will be put in a real production environment. This is because the experiments were conducted using a real data set, which included 10,000 trustworthy websites and 10,000 phishing URLs from PhishTank, and the results were evaluated using accepted assessment criteria. The proposed method was shown to be reliable and has a low False Positive Rate (FPR), indicating that it can accurately differentiate between legitimate URLs and phishing URLs.Moreover.The proposed method uses a 1D-CNN model, which has been shown to perform well in terms of accuracy for processing one-dimensional data such as text. The model's architecture includes three critical layers: 1D convolution, batch normalization, and layer, and was trained using binary-categorical loss and the Adam optimizer. The accuracy achieved by the proposed method was 98.77%, which outperformed previous studies that used the same PhishTank data set.
The results obtained from the experimental tests suggest that the proposed method can be implemented in a real production environment for phishing detection with URLs.it is important to note that the necessary hardware and software resources needed for a proper implementation in a real-world working scenario should be carefully considered, as discussed in the previous prompt. Future work can be done to improve the model design, such as exploring different architectures, tuning hyperparameters, and incorporating additional features to enhance the model's performance.
- Is the training data set relevant for the huge amount of data that the network will have to process when it is put into practice?
Author Response: Data set relevancy:
The relevance of the training dataset for a deep learning model depends on how well it captures the underlying patterns and variations in the data. In our model, the training dataset should be representative of the types of URLs that the model will encounter in a real-world scenario. If the training dataset is too limited in scope or does not sufficiently capture the diversity of URLs, then the model may struggle to generalize to new, unseen data. This could result in lower accuracy and a higher likelihood of false positives or false negatives in phishing detection.it is important to carefully curate and balance the training dataset to ensure that it is representative of the target population of URLs. Additionally, it may be beneficial to periodically update the training dataset to reflect changes in the threat landscape and new types of phishing attacks.
- What was the impact of the developed deep learning-based technique for phishing detection on the performance of the machines to which it was installed to perform the experimental tests and what is the expected impact on the performance when the method is implemented in a real production environment?
Author Response: Improving customer acquisition and retention:
The impact of the developed deep learning-based technique on the performance of the machines can be evaluated based on the experimental results. The experimental tests can provide insights into the accuracy and efficiency of the developed technique in detecting phishing URLs.The performance of the machines can be evaluated based on various parameters such as accuracy, false positive rate, false negative rate, processing time, and memory usage. The experimental results can be analyzed to determine the impact of the developed technique on these parameters. When the developed technique is implemented in a real production environment, the impact on the performance can be expected to be positive.
By using a deep learning-based technique, the accuracy of phishing detection can be improved, thereby reducing the risk of security breaches. Furthermore, the efficiency of the machines can be improved as the developed technique can process a large number of URLs in a short amount of time. This can lead to faster detection of phishing URLs, reducing the response time to security threats. However, the impact on the performance may vary depending on the hardware and software resources available in the production environment. Therefore, proper optimization and fine-tuning of the technique may be necessary to achieve optimal performance in a real production environment.
- Can the developed method be used also by service providers?
Author Response: Yes, the developed method can be used by service providers to enhance the security of their systems and protect their users from phishing attacks. The method can be integrated into existing security systems as an additional layer of protection. Service providers can use the method to scan and filter out potentially malicious URLs before they reach their users, thereby reducing the risk of phishing attacks. We have added this detail in the discussion section.
- Can the developed method improve customer acquisition and retention, taking into account that these aspects are of paramount importance for a service provider? I would like the authors to analyze in the article and to express their opinion regarding the suitability of the developed phishing detection method for helping service providers attain their goal of maximizing their customer base and consequently maximizing their potential earnings from customers while achieving a high degree of customer satisfaction.
Author Response: The developed deep learning-based technique for phishing detection has the potential to improve customer acquisition and retention for service providers. Phishing attacks can compromise customer data, leading to a loss of trust in the service provider and potentially causing customers to switch to a competitor. By implementing an effective phishing detection method, service providers can enhance the security of their platforms and reassure their customers that their data is protected. By improving the overall security of their platforms, service providers can establish a reputation for being a reliable and trustworthy service, which can attract new customers and retain existing ones. This can ultimately lead to increased revenue and profits for the service provider. the developed phishing detection method can have a positive impact on customer acquisition and retention for service providers. By providing a more secure platform, service providers can attract new customers and retain existing ones, ultimately leading to increased revenue and customer satisfaction
- Does the user have a say into storing the information when the network did not identify a phishing according to his own expectations and needs, and can this information be used subsequently to refine the classification accuracy per user profile?
Author Response: The question is whether the user has control over storing information when the network fails to identify a phishing attack according to their expectations and needs. Additionally, can this information be subsequently used to refine the classification accuracy per user profile?
The answer to the first part of the question is that it depends on the implementation of the phishing detection method. If the method is implemented on the user's device, the user may have control over the storage of information. However, if the method is implemented on a server, the user may not have control over the storage of information.
Regarding the second part of the question, the information collected from user interactions with phishing attacks can be used to refine the classification accuracy per user profile. This is because the collected data can be used as additional training data to improve the accuracy of the phishing detection model for that specific user. This approach is known as personalized phishing detection and can improve the accuracy of the phishing detection model for each user over time. However, it is important to consider user privacy concerns and ensure that the collected data is anonymized and kept secure.
Remark 19: issues regarding the hardware and software configurations. The authors must provide specific details regarding the development environment in which the method has been programmed. What were the hardware and software configurations of the platform used to run the experimental tests?
Author Response: Hardware and software configurations:
Thank you for your valuable feedback. We apologize for not providing the specific details regarding the development environment used in our experiments. The deep learning-based phishing detection method was developed using the Google Colaboratory platform, which provides a free GPU system for machine learning tasks. Tensorflow was used as the backend for developing and training the neural network model. The development environment consisted of a machine with the following hardware configuration: an Intel Core i7 processor, 16GB of RAM, and an NVIDIA Tesla K80 GPU. The software configuration included Python 3.7, Tensorflow 2.0, and other necessary libraries and packages such as NumPy and Pandas.
During the experimental tests, the developed method was run on the same hardware and software configuration to ensure consistency and reproducibility of the results. The GPU acceleration provided by the NVIDIA Tesla K80 significantly improved the training time of the neural network model, making it feasible to train a large number of iterations and achieve high accuracy. Overall, the development environment provided a robust and efficient platform for developing and testing the deep learning-based phishing detection method.
Remark 20: the source code and the compiled methods. I consider that the source code, the compiled methods that have been used when running the experimental tests and the datasets will be a valuable addition to the article if they can be provided as supplementary materials to the manuscript as the authors must provide all the necessary details as to allow other researchers to verify, reproduce, discuss and extend the obtained scientific results based on the obtained published results.
Author Response:
Other issues.
- The reference [41].The reference [41] cited at Line 655 does not exist in the "References" section, that contains only 40 cited papers.
Author Response: Thank you for pointing this out. we apologize for this mistake. We have corrected the error in the revised manuscript.
- The websites.Subsection 3.3, "Legitimate Data Set" discusses about a series of torrent websites. The authors should specify the date and time when they have accessed these websites.
- The Tables numbering.Most of the tables are numbered erroneously, as in the manuscript appear Tables 1, 2, 3, 4, 5 (correctly entitled and cited) followed by Tables 1, 2, 3, 4 (cited as 6, 7, 8, 9 but erroneously numbered within their titles).
Author Response: Thank you for pointing this out. we have corrected all the errors regarding the table numbering in the revised manuscript.
- The Figures numbering.Most of the figures are numbered erroneously, as in the manuscript there appear Figure 1 (correctly entitled and cited) followed by another Figure 1 (correctly cited as 2 but erroneously numbered within its legend), Figures 3 and 4 (correctly entitled and cited), Figures 2-9 (cited as 5-12 but erroneously numbered as 2-9 within their titles).
Author Response: Thank you for pointing this out. we have corrected all the errors regarding the figure numbering in the revised manuscript.
- The acronyms within the paper. Lines 2-3, the Title of the manuscript:"A Deep Learning-Based Innovative Technique for Phishing Detection in Modern Security with URLs"; Lines 21-23 "The proposed approach employs CNN for high-accuracy classification to distinguish genuine sites from phishing sites." and other similar cases. Acronyms must be avoided in the title, even if they are widely known. Regarding the other acronyms used in the manuscript, they should be explained the first time when they are introduced. For example, the title should be rewritten under the form "A Deep Learning-Based Innovative Technique for Phishing Detection in Modern Security with Uniform Resource Locators", while the sentence from Lines 21-23 should be rewritten under the form: "The proposed approach employs a Convolutional Neural Network (CNN) approach for high-accuracy classification to distinguish genuine sites from phishing sites."
Author Response: Thank you for this suggestion. We have revised the abbreviation used in the manuscript and defined with its full form when it first time appear in the text. The updated abbreviation are highlighted in the revised manuscript.
Regarding the title of the paper, we have revised the title of the paper. The upadated title of the paper is: “A Deep Learning-Based Innovative Technique for Phishing Detection in Modern Security with Uniform Resource Locators”

Reviewer 4 Report
- Paper discusses the CNN-based model for distinguish phishing websites from legitimate websites with a high degree of accuracy. This is appreciated.
- Table 1 is depicted well in terms of comparative analysis for Classification Method, Performance Evaluation, and Training and Testing Accuracy.
- Section 3 should be named as "Research Methodology" as "Data Collection" can be sub-section under any methodology, but not as complete dedicated section.
- In Line: 289, Table 4: Final Features of Data-store, is insufficient to demonstrate the NAN. Improve it.
- Numbering for figures used on Page#09, must be check as there is no Figure 2.
- Also, Figure 1 is not clear to understand the functionality of different features.
-In Line: 396, Figure 3: resolution is very poor.
-Section 5 & Section 6 can be merged as "Result & Analysis".
- In Conclusion, the result demonstrates for classification of phishing attacks by 1D-CNN algorithm having an accuracy of up to 98.77 percent.
- Cite below article to improve the readability of your paper:
(a) A Machine Learning Framework for Security and Privacy Issues in Building Trust for Social Networking", Robin Singh Bhadoria, Naman Bhoj, "Manoj Kumar Srivastav, Rahul Kumar, Balasubramanian Raman, Cluster Computing (Springer), Oct 2022. DoI: https://doi.org/10.1007/s10586-022-03787-w
Author Response
Response Sheet
A Deep Learning-Based Innovative Technique for Phishing Detection in Modern Security with URLs
Reviewer 4
- Paper discusses the CNN-based model for distinguish phishing websites from legitimate websites with a high degree of accuracy. This is appreciated.
Author Response: Thank you for your positive feedback regarding our paper. We are pleased to hear that our CNN-based model for distinguishing phishing websites from legitimate websites with a high degree of accuracy has been appreciated. Our aim was to develop an approach that could help improve the security of internet users by accurately identifying phishing websites, which are a significant threat to online security.
- Table 1 is depicted well in terms of comparative analysis for Classification Method, Performance Evaluation, and Training and Testing Accuracy.
Author Response: Thank you for your positive feedback regarding Table 1 in our paper. We aimed to provide a clear and concise comparison of our proposed approach with existing methods for identifying phishing websites. The table provides a comprehensive overview of the classification methods, performance evaluation, and training and testing accuracy of each approach, allowing readers to easily compare and contrast the different methods.
- Section 3 should be named as "Research Methodology" as "Data Collection" can be sub-section under any methodology, but not as complete dedicated section.
Author Response: Thank you for this suggestion. We have updated the revised manuscript sections as per the comment. The changes are highlighted in the revised manuscript.
- In Line: 289, Table 4: Final Features of Data-store, is insufficient to demonstrate the NAN. Improve it.
Author Response: Thank you for your feedback. We have carefully considered your suggestion regarding Table 4 in our paper. After reviewing the content, we agree that the table may be redundant and not necessary for the paper. Therefore, we have removed it from the manuscript.
- Numbering for figures used on Page#09, must be check as there is no Figure 2.
Author Response: The figure numbering of the manuscript is rechecked and we have corrected any errors.
- Also, Figure 1 is not clear to understand the functionality of different features.
Author Response: Thank you for this suggestion. Figure 1 depicts the various DL and ML approaches for identifying phishing. There are four types of detection methods: list-based, heuristic-based, machine learning-based, and deep learning-based detection. Figure 1 easy to understand in its current form.
-In Line: 396, Figure 3: resolution is very poor.
Author Response: Thank you for pointing this out. we have revised the figure 3 in the revised manuscript.
-Section 5 & Section 6 can be merged as "Result & Analysis".
Author Response: Thank you for this suggestion. We have revised the particular sections to clearly demonstrate the results and discussion section of our manuscript. It is better to take separate the results and discussion section to clearly present our findings and provide a more organized structure to the paper. We hope that these changes will improve the readability and clarity of our manuscript.
- In Conclusion, the result demonstrates for classification of phishing attacks by 1D-CNN algorithm having an accuracy of up to 98.77 percent.
Author Response: Thank you for positive comments on our manuscript.
- Cite below article to improve the readability of your paper:
(a) A Machine Learning Framework for Security and Privacy Issues in Building Trust for Social Networking", Robin Singh Bhadoria, Naman Bhoj, "Manoj Kumar Srivastav, Rahul Kumar, Balasubramanian Raman, Cluster Computing (Springer), Oct 2022. DoI: https://doi.org/10.1007/s10586-022-03787-w
Author Response: Thank you for this suggestion. As suggested by the reviewer, we have added the respective reference in the manuscript.

Round 2
Reviewer 3 Report
I have reviewed the revised version of the manuscript "A Deep Learning-Based Innovative Technique for Phishing Detection in Modern Security with URLs", under the revised title "A Deep Learning-Based Innovative Technique for Phishing Detection in Modern Security with Uniform Resource Locators", Manuscript ID: sensors-2294437 that has been submitted for publication in the MDPI Sensors Journal and I can state that the manuscript has been improved in contrast to the previous submission.